# Building Archimedean Space

Bogdan Stoica

*Martin A. Fisher School of Physics, Brandeis University, Waltham, MA 02453, USA*

*Department of Physics, Brown University, Providence RI 02912, USA*

bstoica@brandeis.edu

## Abstract

I propose that physical theories defined over finite places (including $p$-adic fields) can be used to construct conventional theories over the reals, or conversely, that certain theories over the reals "decompose" over the finite places, and that this decomposition applies to quantum mechanics, field theory, gravity, and string theory, in both Euclidean and Lorentzian signatures. I present two examples of the decomposition: quantum mechanics of a free particle, and Euclidean two-dimensional Einstein gravity. For the free particle, the finite place theory is the usual free particle $p$-adic quantum mechanics, with the Hamiltonian obtained from the real one by replacing the usual derivatives with Vladimirov derivatives, and numerical coefficients with $p$-adic norms. For Euclidean two-dimensional gravity, the finite place objects mimicking the role of the spacetime are $\mathrm{SL}_2(\mathbb{Q}_p)$ Bruhat-Tits trees. I furthermore propose quadratic extension Bruhat-Tits trees as the finite place objects into which Lorentzian $\mathrm{AdS}_2$ decomposes, and Bruhat-Tits buildings as a natural generalization to higher dimensions, with the same symmetry group on the finite and real sides for the manifolds and buildings corresponding to the vacuum state. I comment on the implications of this decomposition for the cosmological constant problem, black hole information loss problem, and construction of black hole microstates.

BRX-TH-6329, Brown-HET-1763

# 1 Introduction

The idea that number theory should play a role in physics is not new. There exists a certain point of view from which this suggestion is well motivated: if prime numbers are in some sense the fundamental building blocks of number theory, it is natural to expect that they make an appearance in physics as well, should one work at a deep enough level. Over the years, there have been numerous papers either advocating in this direction, or attempting to construct physical theories out of number theoretic objects.[1] Manin [1] proposed that physics over the reals should be recoverable from adelic physics, and the $p$-adic program of the '80s and early '90s succeeded in constructing some $p$-adic analogues of quantum mechanical systems [2–10] and of conformal field theories [11]. On the string theory side, [12–22] constructed $p$-adic strings, mostly focusing on amplitudes and related machinery. Zabrodin [23, 24] developed a non-Archimedean analogue of the open string worldsheet, which can also be interpreted as a simple example of $p$-adic AdS/CFT, although it was not recognized as such at the time. Further possible forms for scattering amplitudes were worked out by [25–28], and hierarchical models, together with renormalization groups and powerful non-renormalization results, were studied in [29–35]. Manin and Marcolli [36] connected holography to Arakelov geometry, building on earlier work by Manin [37, 38].[2]

Over the last couple of years, the subject of number theory in physics has seen renewed interest, motivated in part by $p$-adic AdS/CFT [43] and by analogies between tensor networks and Bruhat-Tits trees [44]; other recent works include [45–54]. However, it is safe to assert that despite all this activity, up to the present moment the connection between number theory and the kinds of physics relevant for the world we live in remains elusive.

In this paper, I would like to take some steps toward connecting non-Archimedean physics to the more usual fare of contemporary high-energy theory research. The deep motivation for this is attempting to identify the microscopic degrees of freedom in quantum gravity, but it will turn out that the tools we will come across ($p$-adic decomposition and reconstruction) can apply to other types of theories as well. Although in this paper I will only discuss gravity and quantum mechanics, given results already

---

[1]Almost any review of a sector of the physics literature is bound to omit some important works. In the present case, this should be interpreted as coming from a lack of familiarity of the author with the literature, and not from any shortcomings of the omitted works.

[2]Other papers related to the study of non-Archimedean physics include [39–42].

existent in the $p$-adic literature, it is immediate to conjecture that $p$-adic decomposition and reconstruction should also apply to field and string theories.[3] This should not be surprising: the distinction between gravitational and non-gravitational physics is arbitrary, so it should be expected that something akin to a reformulation of gravity will also apply to other types of theories. From this point of view, it is natural to ask whether the techniques introduced in this paper could also help define field and string theories in mathematically rigorous ways.

The proposal will have certain attractive features, on which we now comment:

1. Archimedean locality gets scrambled at the finite places (and different finite places are scrambled from each other). In the quantum mechanics of Section 2, this will happen at two levels: (i) The norm over the $p$-adic numbers is different from that over the reals, and also (ii) The Vladimirov derivative, which enters the $p$-adic Schrödinger equation, is a nonlocal operator. Nevertheless, locality will be neatly restored when the Archimedean side is reconstructed.

2. Ordinary derivatives are not present in the theory. It is an often stated piece of lore that a theory of quantum gravity should not have derivatives, since the arbitrary closeness of the two points in the derivative clashes with the Planck scale.

3. Unitful scales decouple from the dynamics. This is most evident in field theories (not discussed in this paper), but it will also be a feature, in a less pronounced manner, of the quantum mechanics of Section 2. This decoupling of units does not imply that $p$-adic field theories will exhibit no RG running; rather, there should be deep connections between the $p$-adic side and Archimedean RG.

It is important to emphasize that although modern high energy physics is currently mostly not number theoretic in nature, hints of hidden number theoretic structures abound. Rather than being accidental, the point of view advocated in this paper suggests that this is natural. If the Archimedean theories arise from $p$-adic ones, the objects through which the connection is made must have appropriate number-theoretic forms.

Let's now quickly comment on the proposal: Archimedean theories can be constructed from non-Archimedean theories, and conversely Archimedean theories decompose over the finite places. There is a slight abuse of terminology here, in that the

---

[3]Informally, we can think of the $p$-adics as a "layer" at which theories can be defined, independent of the other (quantum mechanical, field theoretic, stringy) layers in physics.

Archimedean theories obtained from $p$-adic reconstruction are not quite the same as the theories that would be defined directly on the Archimedean side without any mention of $p$-adics, and indeed there is no totalitarian principle that they should be exactly the same. For our choice of examples, some of these differences will be spelled in detail in Sections 2 and 4. Another important point is that, in general, there could also be an adelic component to the reconstruction, or said another way, the finite places may not be sufficient to reconstruct the Archimedean theory. This does not happen for the simple examples in Sections 2 and 4, but it could occur generally. The reader should thus keep in mind that the constructions in this paper are secretly adelic, although for most of the discussion this will be pushed into the background, and only brought up occasionally. From the adelic point of view, the present paper can be thought of as expanding the proposal in Manin's essay [1].

I should also remark on the connection of the present paper to string theory and loop quantum gravity. Since for $AdS_2$ space the associated building is the Bruhat-Tits tree, it may be tempting to think that once curvature and edge length dynamics are introduced on the tree, one should look for a discretization of $AdS_2$ that assigns patches of space to the edges and nodes in the tree, and that such a discretization could even be performed in the higher-dimensional cases. However, this cannot be right: discretizations of Archimedean space break diffeomorphism invariance, and this breaking is an undesirable feature for many purposes, from which it is not possible to recover. Rather, it should be emphasized that the tree naturally lives at a finite place, and not at the Archimedean one, so that there is no natural embedding of the tree into $AdS_2$.[4] Contact with the Archimedean place is made through functions which take all places into account; for instance, the Archimedean partition function will be related to the finite place partition functions through an Euler product.

Comparing with string theory, the situation is different. There exists a theory of $p$-adic open strings, and the various forms of scattering amplitudes that have already been worked out in the literature suggest that $p$-adic analogues of superstrings should also exist. In fact, string theory will almost certainly be required if a $p$-adic version of interaction unification is to exist.[5] In some sense, the $p$-adic direction is thus orthogonal

---

[4]Nonetheless, it is important to leave open the possibility that more involved constructions across places, which single out special regions of the Archimedean spacetime in a diffeomorphism invariant way, could exist. I thank M. Marcolli for this point.

[5]The Ricci curvature construction of [56–58] that has been used in [45] to obtain edge-length dynamics also admits higher neighbor versions, and it may be natural to interpret these terms as higher

to the $\alpha'$ correction direction of string theory.

The summary of the paper is as follows. In Section 2 we cover the $p$-adic free particle, and its relation to its Archimedean cousin. In Section 3 we present the general proposal for $p$-adic reconstruction, give some dictionary entries between the $p$-adic and Archimedean sides, and comment on the possible implications of $p$-adic decomposition for the microstates of quantum gravity. Section 4 discusses the reconstruction of two-dimensional Euclidean gravity, and Section 5 takes a few steps toward reconstructing Lorentzian two-dimensional gravity. We close in Section 6 with a discussion of some possible consequences for the cosmological constant problem, and for the black hole information loss problem.

The reader should beware that most mathematical review has been relegated to the appendices. To ease parsing through the paper, here is an informal glossary of some terms that will be used:

**Archimedean place:** The continuum spaces over which real-world physics is defined. Can refer to $\mathbb{R}$, anti-de Sitter spaces, etc. Also known as the place at infinity.

**Non-Archimedean place:** "Space" that is not Archimedean. Can refer to the $p$-adic field $\mathbb{Q}_p$, to a finite field $\mathbb{F}_p$, but also to Bruhat-Tits trees and buildings, Drinfeld spaces, etc. Also known as finite place.

**The $p$-adic field $\mathbb{Q}_p$:** A non-Archimedean field defined as the completion of $\mathbb{Q}$ with respect to the $p$-adic metric $|\cdot|_{(p)}$ given by $|x|_{(p)} := p^{-n}$ if $x = p^n a/b$ and $a$ and $b$ do not contain any powers of prime $p$, for integers $a, b$ and rational $x$. $\mathbb{Q}_p$ contains $\mathbb{Q}$, the field of rational numbers. Although $p$-adic strictly refers to $\mathbb{Q}_p$, throughout the paper "$p$-adic decomposition" and "reconstruction" will sometimes loosely mean non-Archimedean decomposition and reconstruction. This should be apparent from the context.

**Bruhat-Tits tree:** An infinite tree of uniform valence at all vertices. Its boundary is $P^1(\mathbb{Q}_p)$, or a projective space defined on an extension of $\mathbb{Q}_p$. In cases when the tree is defined from $\mathbb{Q}_p$, the valence at all vertices is $p + 1$.

# 2   $p$-adic quantum mechanics revisited

In this section I revisit $p$-adic quantum mechanics. My starting point will be some of the results of [2–10], so in this sense Sections 2.1 and 2.2 below can act as review, but

---

curvature corrections, some suitable generalizations of which could be organized as $\alpha'$ expansions.

at the same time I will extend and modify these results, and my point of view will be different.

The guiding principle of at least some of the authors [2–10] was that simple quantum mechanical systems, such as the harmonic oscillator, exhibit a certain $SL_2$ symmetry, which allows the Weyl approach to work in the same way in the classical, quantum and (classical and quantum) $p$-adic cases. This approach is based on a triple $(L^2(\mathbb{Q}_p), W(z), U(t))$, where $L^2(\mathbb{Q}_p)$ (the space of square integrable functions on $\mathbb{Q}_p$) is the Hilbert space, $z$ is a point in the classical phase space, $W(z)$ is the Weyl representation of the commutation relations and $U(t)$ is a time evolution operator. For more details see e.g. [2], and [7] for the special place $p = 2$. This construction of $p$-adic quantum mechanics admits a path integral, developed by Zelenov [8].

In this paper I will adopt the triple structure described above, but in addition to this data I will equip the quantum mechanics with a square-free parameter $\tau \in \mathbb{Q}_p$, specifying a quadratic extension $\mathbb{Q}_p[\sqrt{\tau}]$. The reason for introducing this additional parameter is that in more general contexts (see e.g. [50], and Section 5 below) certain nontrivial values of $\tau$ are intimately connected with time evolution that resembles Archimedean Lorentzian time evolution. From this point of view, the works [2–10] correspond to $\tau = 1$. Since quantum mechanics is non-relativistic, we will not see a striking difference between the $\tau = 1$ and $\tau \neq 1$ cases, however there will still be some differences, so it is best to keep $\tau$ nontrivial. It is also important to emphasize that despite the introduction of this parameter, the wavefunctions $\psi$ (at some fixed time $t$) remain defined as[6]

$$\psi : \mathbb{Q}_p \to \mathbb{C}. \tag{2.1}$$

The reason quadratic extensions are important is that they allow the introduction of nontrivial sign functions on $\mathbb{Q}_p$. For $p > 2$, fixed $\tau$ and $\epsilon$ a primitive $(p^2 - 1)$-th root of unity, there are four equivalence classes of quadratic extensions, with representatives

$$\{\mathbb{Q}_p, \mathbb{Q}_p[\sqrt{\epsilon}], \mathbb{Q}_p[\sqrt{p}], \mathbb{Q}_p[\sqrt{\epsilon p}]\}, \tag{2.2}$$

and thus four sign functions $\text{sgn}_\tau$, one of which is always trivial. At the place $p = 2$ there are 8 such representatives; see Appendix C.3.5 for more details. Given equivalence classes (2.2) above, a fixed $\tau \in \mathbb{Q}$ will fall in different classes at different places $p$. An

---

[6]It is of course possible to attempt defining quantum mechanics on quadratic or higher extensions of $\mathbb{Q}_p$, but I will not do so here.

important difference between the Archimedean and non-Archimedean sign functions is that $\mathrm{sgn}_\tau(-1)$ is not always negative.[7]

Once sign functions $\mathrm{sgn}_\tau : \mathbb{Q}_p^\times \to \{\pm 1\}$ are introduced, it is possible to define multiplicative characters of $\mathbb{Q}_p^\times$ as

$$\pi_{s,\tau}(x) := |x|^s \, \mathrm{sgn}_\tau x, \qquad (2.3)$$

such that $\pi_{s,\tau}(x_1)\pi_{s,\tau}(x_2) = \pi_{s,\tau}(x_1 x_2)$. Additive characters $\chi : \mathbb{Q}_p \to \mathbb{C}^\times$ also exist,

$$\chi(x) := e^{2\pi i \{x\}_p}, \qquad (2.4)$$

such that $\chi(x_1)\chi(x_2) = \chi(x_1 + x_2)$. Here the fractional part $\{x\}_p$ is defined by dropping the integer part of $x$, i.e. the non-negative powers of $p$ in the $p$-adic expansion of $x$ (Eq. (B.4)). The additive characters are nothing more than the analogues of the Archimedean complex exponential; with them, the direct and inverse Fourier transforms can be defined just as in the Archimedean case,

$$F(\omega) = \int F(t)\chi(\omega t), \qquad (2.5)$$

$$F(t) = \int F(\omega)\chi(-\omega t). \qquad (2.6)$$

More details on the mathematical machinery discussed up to this point can be found in Appendices A, B, C.

## 2.1 The Vladimirov derivative

In other to define Hamiltonians for $p$-adic quantum mechanics a notion of derivative must be introduced. This notion is the so-called Vladimirov derivative, which is an integral (non-local) operator.

The Vladimirov derivative $\partial_x^s$, $s > 0$, acting on a function $\psi$ was originally defined as multiplication by $|k|^s$ on the Fourier transform of $\psi$, with $k$ the Fourier space variable (see e.g. [10]). The result can then be Fourier transformed back to position space, which often involves regularization. Different expressions exist in the literature for the position space result, corresponding to different ways of dealing with the regularization.

---

[7]In fact, this is also true at the Archimedean place if one considers the trivial sign function $\mathrm{sgn}_1 x = 1$, $\forall \, x \in \mathbb{R}^\times$.

In the present paper I will take the point of view that the Vladimirov derivative can be more naturally associated to a multiplicative character; this was already suggested by [50], and can also be understood from [55]. For $s \neq -1, 0$, the Fourier space definition of the (position space) Vladimirov derivative thus is

$$\partial_k^{s,\tau} \psi(k) = \pi_{s,\tau}(k)\psi(k), \tag{2.7}$$

where $\pi_{s,\tau}(k)$ is a multiplicative character as in Eq. (2.3).[8] The position space action of the Vladimirov derivative is given by Fourier transforming,

$$\partial_x^{s,\tau} \psi(x) = \int \pi_{s,\tau}(k)\psi(k)\chi(-kx), \tag{2.8}$$

where

$$\psi(k) = \int \chi(kx')\psi(x'). \tag{2.9}$$

Performing the $k$ integral in Eqs. (2.7) – (2.8) with the help of Eq. (2.13) below gives the following position space representation for the Vladimirov derivative,

$$\partial_x^{s,\tau} \psi(x) = \Gamma\left(\pi_{s+1,\tau}\right) \int \frac{\psi(x') \operatorname{sgn}_\tau(x' - x)}{|x' - x|^{s+1}}. \tag{2.10}$$

This expression agrees with the formulas given e.g. in [4] for $\tau = 1$, up to regularization and some conventions. Note that there are two (related) levels of regularization at play here: (1) the integral in the Fourier transform of the multiplicative character in Eq. (2.13) is not always convergent, in which case the integral can be defined by analytic continuation instead, and (2) Eq. (2.10) applied to simple functions does not always give a finite result. To regularize this second source of divergences Vladimirov and Volovich proposed an alternate position space expression for the derivative, which for trivial sign character is proportional to (see e.g. [4], [10])

$$\int \frac{\psi(x') - \psi(x)}{|x' - x|^{s+1}}. \tag{2.11}$$

For the purpose of this paper, it is sufficient to work with the formal expression in Eq. (2.10), and so I will not adopt the definition in Eq. (2.11). More details on the Vladimirov derivative can be found in [10].

---

[8]I will sometimes write $\partial^s$ instead of $\partial^{s,\tau}$ when $\tau = 1$.

It is easiest to gain some intuition on the Vladimirov derivative by letting it act on a simple function, say $\chi(Kx)$. We have

$$\partial_x^{s,\tau}\chi(Kx) = \int \pi_{s,\tau}(k)\chi(Kx' + kx' - kx). \tag{2.12}$$

We can perform the $k$ integral first, using Eq. (B.17) in the appendices for the Fourier transform of a multiplicative character, which states that

$$\int \chi(kx)\pi_{s,\tau}(k) = \Gamma\left(\pi_{s+1,\tau}\right)\pi_{-s-1,\tau}(x), \tag{2.13}$$

with the Gamma function defined in Appendix B.3, so that

$$\partial_x^{s,\tau}\chi(Kx) = \Gamma\left(\pi_{s+1,\tau}\right)\int \pi_{-s-1,\tau}(x')\chi\left[K(x'+x)\right]. \tag{2.14}$$

Then one more integration gives

$$\partial_x^{s,\tau}\chi(Kx) = \operatorname{sgn}_\tau(-1)\pi_{s,\tau}(K)\chi(Kx), \tag{2.15}$$

where we have used the Gamma functional equation (B.16).

Let me also briefly comment on the regularization and sources of divergences in Eq. (2.13). The $\mathbb{Q}_p$ integral in this equation can be split into two integrals, over regions $\mathbb{Z}_p$ and $\mathbb{Q}_p - \mathbb{Z}_p$. Depending on the values of $s$ and $\tau$, in general one of these integrals converges, but the other does not. A regularized result for each integral can then be obtained by analytically continuing the result from the convergent region, and the regularized result on the right-hand side of Eq. (2.13) is obtained by summing up the two regularized contributions. The right-hand side of Eq. (2.13) is now regular everywhere, except at $s = 0, \tau = 1$, where it is a representation of the $p$-adic Dirac delta function.

## 2.2 $p$-adic Schrödinger equation and the free particle

In this section we consider the $p$-adic Schrodinger equation, applied to the free particle. Since for Vladimirov derivatives the parameter controlling the order of the derivative doesn't have to be a positive integer, the most general form of the $p$-adic Schrödinger

equation that one could write down is

$$\left(\partial_x^{s',\tau'} + V(|x|)\right)\psi = \mathcal{C}\partial_t^{s,\tau}\psi. \tag{2.16}$$

Here the left-hand side is proportional to the Hamiltonian, $\mathcal{C}$ is a constant that also accounts for parameters such as mass that could appear in the kinetic term, and the partials denote Vladimirov derivatives, with the superscripts referring to the parameters of the multiplicative characters associated to the derivatives, as in Eqs. (2.7) and (2.8). Position $x$ and time $t$ are $p$-adic valued, and the wavefunction is defined on the $p$-adics taking values in the complexes.

For $\tau = \tau' = 1$ the general form (2.16) of the Schrödinger equation has already been studied in the literature, see e.g. [10]. However, my aim here is not to study the most general form of the $p$-adic Schrödinger equation, but rather to match against the usual one-dimensional Archimedean Schrödinger equation, in a sense that will be made precise soon. For this purpose, it suffices to consider equations of the form

$$\left(\frac{1}{|2m|}\partial_x^2 - V(|x|)\right)\psi = \mathrm{sgn}_\tau(-1)\partial_t^{1,\tau}\psi. \tag{2.17}$$

In this equation $|\cdot|$ refers to the $p$-adic norm. Since the mass $m$ is a dimensionful parameter, this expression needs to be interpreted as follows. Nominally, a $p$-adic Planck constant $h$ is also present in Eq. (2.17). If this constant is restored, it is possible to make a dimensionless product of $h$, $m$, and of $x$ and $t$ (coming from the Vladimirov derivatives, cf. Eq. (2.10)) appear inside the $p$-adic norm, so that the norm becomes applied to a unitless number is thus well-defined mathematically. However, in the rest of the paper I will set Planck's constant to unity and work with dimensionless $x$, $t$, $m$, etc. Finally, I should also remark that the $\mathrm{sgn}_\tau(-1)$ coefficient on the right-hand side may appear strange, however it is the correct coefficient to match against the Archimedean side.

For the free particle the Hamiltonian is just a kinetic term,

$$H = \frac{1}{|2m|}\partial_x^2. \tag{2.18}$$

The mass parameter $m$ is $p$-adic, but in order to make contact with the Archimedean place I will demand $m \in \mathbb{Q}$. Using Eq. (2.15) it is immediate to check that plane waves

of the form

$$\psi_{\text{plane}}(x,t) = \chi(kx + \omega t) \tag{2.19}$$

are solutions to the free particle Schrödinger equation, provided that

$$|\omega| \, \text{sgn}_\tau \, \omega = \frac{|k|^2}{|2m|}. \tag{2.20}$$

This equation implies that $\text{sgn}_\tau \, \omega = 1$, and, just as for the mass $m$, in order to make contact with the Archimedean place I will demand $k, \omega \in \mathbb{Q}$, although a priori $k$ and $\omega$ could be $p$-adic parameters.

As usual, the Schrödinger equation can be explicitated either in position or momentum space, and Eqns. (2.16) – (2.18) up to this point have been in position space. Going to momentum space, the free particle Schrödinger equation reads

$$\left| \frac{k^2}{2m} \right| \psi(k,t) = \text{sgn}_\tau(-1) \partial_t^{1,\tau} \psi(k,t) \tag{2.21}$$

and the free particle propagator is given by

$$K(k,t) = \chi\left( \frac{k^2 t}{2m} \right), \tag{2.22}$$

so that the wavefunction at any ($p$-adic) time $t$ can be constructed as

$$\psi(k,t) = K(k,t)\psi(k,0). \tag{2.23}$$

Note that propagator (2.22) satisfies the Schrödinger equation only if

$$\text{sgn}_\tau(2m) = 1. \tag{2.24}$$

For $\tau = 1$ this condition is trivial, but for arbitrary $\tau$ it will place nontrivial restrictions on the allowed values of the mass. The mass $m$ being positive at the Archimedean place does not imply $\text{sgn}_\tau(2m) = 1$ at the finite places. However, it is possible to work with masses that are positive at all places; I will come back to this point in Section 2.3.

The position space propagator can be obtained by Fourier transforming,

$$K(x,x',t) = \int K(k,t)\chi\left[k(x - x')\right]. \tag{2.25}$$

The $k$ integral can be performed with the help of the Gaussian integral (B.12), and the position space propagator at all finite places $p$ equals (see Ref. [2, 7, 10])

$$K(x, x', t) = \lambda \left( \frac{t}{2m} \right) \left| \frac{m}{2t} \right|^{\frac{1}{2}} \chi \left( -\frac{m(x' - x)^2}{2t} \right),$$

(2.26)

so that the time evolved wavefunction is

$$\psi(x, t) = \int K(x, x', t) \psi(x', 0).$$

(2.27)

The coefficient $\lambda$ is a phase factor which arises out of the Gaussian integral; its precise value is given in Appendix B.2 below. Eqs. (2.26) – (2.27) satisfy the Schrödinger equation (2.17) with $V = 0$ in position space representation, provided that positivity condition (2.24) holds.

Time evolution by propagators (2.22), (2.26) must obey

$$U(t)U(t') = U(t + t').$$

(2.28)

This condition is trivially (multiplicatively) satisfied by the momentum space propagator $K(k, t)$, however it produces a nontrivial self-consistency condition for the position space propagator $K(x, x', t)$ in Eq. (2.26), which implies that the phase factors $\lambda$ must satisfy (for a proof of this relation see e.g. [10])

$$\lambda(a)\lambda(b)\lambda \left( -\frac{a + b}{ab} \right) = \lambda(a + b).$$

(2.29)

In particular, note that Eq. (2.28) does not allow the freedom of multiplying the propagators by arbitrary coefficients.

One difference from the Archimedean case is that even though Eqns. (2.26), (2.27) resemble time evolution, there is no natural notion of ordering defined on the $p$-adics, so there is no immediate notion of time ordering associated to this evolution. In fact, various orderings on $\mathbb{Q}_p$ are possible, such as the linear order of [8], or the ordering of [4], which admits a Cauchy problem interpretation.

## 2.3 Reconstructing the Archimedean free particle

It is finally time to connect to Archimedean quantum mechanics. There are several Archimedean objects one could ask about when reconstructing from the $p$-adic side:

1. Partition functions and propagators

2. Hamiltonians and time evolution

3. Operators and wavefunctions

In this section the sharpest results will be for the propagators and time evolution, with a few results on wavefunction reconstruction also. A more general perspective on Archimedean reconstruction will be given in Section 3.

Of course, operators and wavefunctions are in some sense dual to each other. While from the quantum mechanical point of view of this section wavefunction reconstruction may be a natural question, a more field theoretic point of view may prefer asking about operators, with the wavefunctions fixed to some reference (vacuum or otherwise) state. I will not have much to say about arbitrary operator reconstruction in this paper.

### 2.3.1 Propagators

Introducing the Archimedean additive character

$$\chi_{(\infty)}(x) := e^{-2\pi i x} \tag{2.30}$$

and the Archimedean phase factor[9]

$$\lambda_{(\infty)}(a) := \exp\left(-i\frac{\pi}{4}\operatorname{sgn}_{\tau=-1}^{(\infty)} a\right), \tag{2.31}$$

with $\operatorname{sgn}_{\tau=-1}^{(\infty)}$ the usual sign function on $\mathbb{R}$ (written in the notation of Appendix C), the momentum and position space propagators obey the Euler product formulas

$$K_{(\infty)}(k,t) = \prod_{p=2}^{\infty} \frac{1}{K_{(p)}(k,t)}, \qquad K_{(\infty)}(x,x',t) = \prod_{p=2}^{\infty} \frac{1}{K_{(p)}(x,x',t)}, \tag{2.32}$$

---

[9]This phase factor comes from the Gaussian integral $\int \chi_{(\infty)}\left(ax^2 + bx\right) = \lambda_{(\infty)}(a)|2a|_{(\infty)}^{-\frac{1}{2}}\chi_{(\infty)}\left(-\frac{b^2}{4a}\right)$, the Archimedean analogue of Eq. (B.12).

so that $K_{(\infty)}(k,t)$, $K_{(\infty)}(x,x',t)$ are still given by Eqs. (2.22), (2.26), but with the finite place objects replaced by Archimedean ones. The momentum space formula in Eq. (2.32) comes about because of the Euler product formula (D.4) for the additive characters, while the position space formula occurs because of three separate Euler product identities (D.4), (D.5), (D.7), for the additive character, norm, and phase factor $\lambda$. Position space Archimedean propagator (2.32) solves the Schrödinger equation

$$-\frac{1}{2m}\partial_x^2\psi(x,t) = 2\pi i\partial_t\psi(x,t), \tag{2.33}$$

and the momentum space Archimedean propagator solves equation

$$\frac{k^2}{2m}\psi(k,t) = \frac{i}{2\pi}\partial_t\psi(k,t), \tag{2.34}$$

where the derivatives are now ordinary derivatives and the wavefunctions are defined as $\psi : \mathbb{R} \to \mathbb{C}$, with Hilbert space $L^2(\mathbb{R})$, as usual. Thus, at the Archimedean place $k$ plays the role of momentum, and not of wavenumber; up to $h$, there does not seem to be a difference between momentum and wavenumber at the finite places.

Although formulas (2.32) may not be well-known, they are not new, as the position space result was already noted by [10]. However, what I would like to do in this section is propose a new perspective on these formulas: rather than being accidental, Eqns. (2.32) can be used to *define* time evolution (and, thus, a Hamiltonian) at the Archimedean place, starting purely from $p$-adic data. In this sense, Archimedean time evolution can be seen as emergent from $p$-adic time evolution.[10]

To better understand this proposal, it is important to emphasize that Eq. (2.33) as derived from the $p$-adic propagators cannot make sense for arbitrary real values of the parameters $m$, $k$, $x$, and $t$. This is because arbitrary real numbers are not elements of $\mathbb{Q}_p$, and conversely, arbitrary elements of $\mathbb{Q}_p$ are not elements of $\mathbb{R}$. Rather, the intersection of all $\mathbb{Q}_p$'s and of $\mathbb{R}$ is $\mathbb{Q}$. Thus, Eq. (2.33) can only make sense if all parameters are rationals; this was the reason for demanding rationality back in Section 2.2. Of course, Archimedean physics takes place for real values of all parameters, so we must use the following prescription: if the Archimedean data $(m,x,t)$ we are interested in are rational, then we can use Eq. (2.33) directly; if not, since $\mathbb{Q}$ is dense in $\mathbb{R}$, it

---

[10]The wavefunction in Eq. (2.33) is still Archimedean; wavefunction reconstruction from the finite places will be considered in Section 2.3.2.

is possible to find a rational sequence $(m, x, t)_n$ that obeys Eq. (2.33) and converges to the real values of interest, and the behavior of the irrational data is defined by taking the appropriate limit. Thus, we never write down the equation of motion at irrational points, but this has no impact on the predictivity power of the theory. This type of argument, where irrational points are first excluded and then "patched back in" from density considerations, is a general feature of non-Archimedean constructions of Archimedean physics, and will be revisited in Section 3. For now, it is important to note that it enhances the set of Archimedean theories that can be built from $p$-adic ones, since it allows the $p$-adic data used to be sparse, as long as density at the Archimedean place is still obeyed.

Let's now make a comment on restoring the Planck constant. Inserting a factor of $h$ (not $\hbar$) in the usual spots in Eq. (2.17) will make the same factor of $h$ appear in Eqs. (2.33), (2.34). Thus, the rational quantities are built out of the usual Planck constant and parameters $m, x, t, k$, and do not use the reduced Planck constant.

I now move on to discussing signs. The results we have obtained are functions of the parameter $\tau$ labeling the quadratic extension we have been implicitly using. Setting $\tau = 1$ makes all sign functions trivial, in which case our formulas for propagators and Schrödinger equations match those of the classical literature [2–4, 6, 10] on $p$-adic quantum mechanics. But let's consider what happens if we set $\tau = -1$.[11] For this particular value of the quadratic extension parameter $\mathrm{sgn}_\tau 2 = 1$ at all places, by definition, since $1 + 1 = 2$. Then Eq. (2.24) simplifies to requiring

$$\mathrm{sgn}_{-1} m = 1 \tag{2.35}$$

at all places.[12] It is possible to arrange this, e.g. by demanding that the mass is a rational squared, and the set of such masses is dense in $\mathbb{R}^+$.[13]

Of course, we don't *have* to demand that our theory enforces $\mathrm{sgn}_{-1} m$ anywhere. We

---

[11]A word of warning: for more complicated physical systems the choice $\tau = -1$ may not be sufficient, since it yields $\mathrm{sgn}_\tau^{(p)}(-1) = -1$ only at $p = 2$. In such situations it is instead possible to pick $\tau$ to be a negative integer at the Archimedean place, which will result in more finite places having $\mathrm{sgn}_\tau^{(p)}(-1) = -1$.

[12]This argument is, of course, independent of the value of the numerical coefficient entering the kinetic term.

[13]Proof that the set of squared rationals is dense in $\mathbb{R}^+$: any $x \in \mathbb{Q}^+$ not a square, there exist rational squares $r_{1,2}$ (pick e.g. $x^2$ and $1/x^2$) such that $r_1 < x < r_2$. But for any rational squares $r_1 < r_2$, there exists a rational $r$ such that $\sqrt{r_1} < r < \sqrt{r_2}$, i.e. so that $r_1 < r^2 < r_2$, so $x$ can be sandwiched between successively closer squared rationals. Finally, use that $\mathbb{Q}^+$ is dense in $\mathbb{R}^+$.

could choose to work with $\tau = 1$, or multiply a $\text{sgn}_\tau(2m)$ in the kinetic term, or adjust the propagator with signs, all of which would also remove the sign dependence. But the mass being positive can be regarded as a kind of condition for the UV well-behavedness of the Archimedean theory, as far as it is possible to talk about the ultraviolet of a nonrelativistic theory. By making the choice $\tau = -1$ this UV well-behavedness follows from the propagator being given by Eq. (2.22), without having to impose it as an additional constraint.

### 2.3.2 Wavefunction reconstruction

Let's now turn to wavefunction reconstruction. Compared to the previous subsection on propagators, the results in this section will be less sharp, in that even for a quantum mechanical free particle it is not clear what the most general rules for constructing Archimedean wavefunctions out of $p$-adic ones should be. While Euler product constructions still make sense in some situations, it is likely that the most general prescription is in fact not a product. We will discuss this possibility below.

Let's start by remarking that it is not obvious for which theories we should expect Archimedean wavefunctions to be recoverable from $p$-adic wavefunctions. The reason for this is as follows: the least we can demand from a quantum $p$-adic model of an Archimedean physical system is that there should exist quantum states at the finite places (which we can call $p$-*states*) for all classical configurations, and that the time evolution of any semiclassical configuration should emerge from the $p$-adic time evolution of the $p$-states. In situations in which an Archimedean quantum theory is available we can demand more: Archimedean wavefunctions should be reconstructible from $p$-states, and the time evolution of any Archimedean wavefunction should be recoverable from the time evolution of $p$-states.

However, it is important to emphasize that an Archimedean quantum theory may not be always available. In other words, for arbitrary quantum $p$-adic theories that reduce to Archimedean ones, the resulting Archimedean theories *may not* be quantum theories, that is the quantum description may only make sense on the $p$-adic side, and have no Archimedean counterpart, other than a semiclassical theory. This observation could be important for gravitational theories, and we will come back to it in Section 3. When this happens, there should be some fundamental obstruction to obtaining Archimedean wavefunctions from $p$-adic ones.

Despite this lack of clarity on when $p$-adic reconstruction of the Archimedean wave-functions can be expected in general settings, we can nevertheless push on and say a few things for simple quantum mechanical systems. The first observation is that, at least in some cases, time evolution itself is sufficient to establish a basis for the Archimedean Hilbert space. For instance, for a $p$-adic model of a quantum harmonic oscillator, the eigenstates of the Archimedean Hamiltonian define a countable basis for the Hilbert space $L^2(\mathbb{R})$, with each basis vector square-integrable. Thus, if the $p$-adic time evolution defines Archimedean time evolution, the Archimedean wavefunctions and Hilbert space can be constructed squarely on the Archimedean side, without the need of a direct prescription on how to obtain them from $p$-adic data. This is also true for the free particle Hamiltonian, if we relax the conditions that the basis should be countable and its basis vectors square-integrable.

Let's now move on to directly reconstructing wavefunctions. From the product identity (D.4), it is immediate that $p$-adic plane waves product into Archimedean plane waves,

$$e^{2\pi i(kx+\omega t)} = \prod_{p=2}^{\infty} \chi_{(p)}(kx + \omega t), \tag{2.36}$$

where we should remember positivity condition (2.20) on the sign of $\omega$ that needs to be satisfied in order for $p$-adic plane waves to satisfy the Schrödinger equation. Just as in the propagator discussion, when $\tau = -1$ it is possible to ensure that the sign is positive at all places, e.g. by choosing $\omega$ to take values in the set of squares of rationals, which is dense in $\mathbb{R}^+$. For $\tau = 1$ there are no nontrivial conditions arising from sign functions.

The next simplest situation to discuss occurs in momentum representation. Suppose we have time evolution on the $p$-adic side,

$$\psi_{(p)}(k,t) = K_{(p)}(k,t)\psi_{(p)}(k,0), \tag{2.37}$$

for some initial data $\psi_{(p)}(k,0)$. Because in this representation time evolution is just multiplication by $K_{(p)}(k,t)$, it is immediate that by defining Archimedean initial data[14]

$$\psi_{(\infty)}(k,0) := \prod_{p=2}^{\infty} \frac{1}{\psi_{(p)}(k,0)}, \tag{2.38}$$

---

[14]By "initial data" here I mean data at $t = 0$; at the finite places a priori there need not be any causality relations between $t = 0$ and some rational $t$ such that $t > 0$ on the Archimedean side.

time evolution on the $p$-adic and Archimedean sides will keep Eq. (2.38) obeyed at all times, that is

$$\psi_{(\infty)}(k,t) = \prod_{p=2}^{\infty} \frac{1}{\psi_{(p)}(k,t)}. \qquad (2.39)$$

In this sense Eqs. (2.38), (2.39) define a class of wavefunctions which are compatible with time evolution, for this particular choice of Hamiltonian.

It is important to emphasize that Eqs. (2.38), (2.39) are rather strange from the point of view of Hilbert spaces. The Archimedean wavefunction $\psi_{(\infty)}(k,0)$ can be in the Hilbert space $L_2(\mathbb{R})$, or the individual $p$-adic wavefunctions can be in their corresponding $L_2(\mathbb{Q}_p)$ Hilbert spaces, however Eqs. (2.38), (2.39) imply that it is difficult to ensure both sides to be elements of their respective Hilbert spaces at the same time. Of course, it could also be the case that *neither* side is in its Hilbert space, as in the plane wave example (2.36). Furthermore, wavefunctions can in general have zeroes for certain values of the coordinates, in which case the interpretation of Eqs. (2.38), (2.39) at those points becomes problematic. These difficulties suggest that the most general rule for obtaining an Archimedean wavefunction out of $p$-states is not a product. I will not attempt to characterize in this paper what this most general rule should be.

How about the position space representation? Since we don't know the most general rules for reconstructing Archimedean wavefunctions, let's restrict the discussion to Euler products. Specifically, just as in momentum representation, we can ask about setting up the Archimedean initial data as

$$\psi_{(\infty)}(x,0) := \prod_{p=2}^{\infty} \frac{1}{\psi_{(p)}(x,0)}. \qquad (2.40)$$

In position representation, time evolution is no longer given by multiplication, so Eq. (2.40) being obeyed at some time $t$ is equivalent to

$$\left( \prod_{p=2}^{\infty} \int_{\mathbb{Q}_p} f_{(p)}(x,x',t) \right) \left( \int_{\mathbb{R}} \prod_{p=2}^{\infty} \frac{1}{f_{(p)}(x,x',t)} \right) = 1, \qquad (2.41)$$

where we have defined

$$f_{(p)}(x,x',t) := K_{(p)}(x,x',t)\psi_{(p)}(x',0). \qquad (2.42)$$

Equation (2.41) has an adelic interpretation. An adele ring $\mathbb{A}$ is the mathematical structure obtained when putting all places together, and elements $a \in \mathbb{A}$ are of the form

$$a = \left( a_{(\infty)}, a_{(2)}, a_{(3)}, \ldots \right), \tag{2.43}$$

where the first entry corresponds to the Archimedean place, and the following entries correspond to the finite places, with all but finitely many $a_{(p)}$'s $p$-adic integers (see Appendix D.1 for more details, or [55] for an in-depth treatment). Functions $f : \mathbb{A} \to \mathbb{C}$ are defined component-wise as a collection of $f_{(v)}$'s over all places, and the integral of $f$ on $\mathbb{A}$ is

$$\int_{\mathbb{A}} f = \prod_v \int_{\mathbb{Q}_v} f_{(v)}, \tag{2.44}$$

where $v$ ranges over all places (finite and Archimedean). It is a requirement of the adelic structure that in this equation only finitely many places should contribute nontrivially; we will come back to the physical meaning of this cutoff in Section 3, and in the conclusion.

Equation (2.44) implies that the set of functions $f_{(p)}$ which keep product (2.41) satisfied at all times are precisely the ones for which the adelic integral (2.44) is trivial, with the Archimedean component defined such that

$$\prod_v f_{(v)} = 1. \tag{2.45}$$

In this sense functions $f_{(v)}$ form the "kernel" of the adelic integral (2.44), but this terminology is nonstandard.

# 3 $p$-adic decomposition and Archimedean space

## 3.1 The proposal

I will now take a step back and explain the results in the previous section from a general perspective.[15] The free particle example we've seen already exhibited features

---

[15]The origins of what I am advocating in this section trace to an essay by Manin [1], who was arguing, in the context of string theory, that Archimedean physics should have a dual $p$-adic description. However, the proposal in this paper is more general, in that I am regarding the non-Archimedean side as fundamental, and not on equal footing with the Archimedean side. Furthermore, I extend Manin's proposal away from string theory, to quantum mechanics, field theory, and gravity, and make it more

of the general story, however, at the same time, since it was only a simple quantum mechanical system, some ingredients were not present.

The most important message of this section is that Archimedean theories can arise from non-Archimedean theories. I will call this phenomenon *p-adic decomposition*, which should intuitively signify that the theory "factorizes" over the finite places; conversely, the theories at the finite places *p-adically reconstruct* the Archimedean theory.[16] Of course, only some of the quantities in a theory which $p$-adically decomposes will literally factorize over the finite places, and only in certain regimes; for arbitrary Archimedean quantities, the rules by which they emerge from $p$-adic objects will be more complicated than a simple product.

Let's ask whether $p$-adic decomposition can be *necessary* for the UV completion of Archimedean theories. As already remarked in Section 2.3.2, there is no a priori reason for an Archimedean theory obtained by $p$-adic reconstruction from a quantum $p$-adic theory to itself be quantum. While in many cases it could happen for the Archimedean theory to be quantum (or quantizable, starting from a classical theory obtained by $p$-adic reconstruction), we have to allow for the possibility that at least in some cases the Archimedean theory could have no quantum description. Said another way, the microstates of what one could hope would be the quantum Archimedean theory could turn out to be $p$-adic in nature, with no Archimedean counterpart. It is certainly reasonable to ask whether this happens for gravity, and I will have no definitive answer in this paper.

Let me now say a few more words on gravity. The reason $p$-adic decomposition may be relevant to quantizing gravity is that it provides a novel possibility for what the microstates of gravitational theories could be. Rather than having to deal with quantizing an Archimedean continuum, it may be possible to quantize a $p$-adic system instead, and then to take an appropriate limit to reconstruct the Archimedean system. In this sense gravity would be an effective theory, although not of a kind that has been encountered previously in the physics literature. If the microstates are strictly $p$-adic, it will never be possible to access them with Archimedean techniques.[17] This proposal

---

precise by giving some explicit dictionary entries.

[16]I am using the term "$p$-adic" loosely here, to refer to reconstructions based on $\mathbb{Q}_p$, but also on Bruhat-Tits trees and buildings and other non-Archimedean objects. A more appropriate term may be "non-Archimedean reconstruction."

[17]A partial exception to this statement are supersymmetric localization methods (and related techniques), which do indirectly count microstates. If the $p$-adic proposal in this section is correct, there should be a simple interpretation at the finite places for what the localization methods are doing.

has *nonlocality* as an attractive feature, in that the gravitational degrees of freedom do not live anywhere in Archimedean space, but rather at the finite places.

It is possible to restate the discussion in the paragraph above in terms of diffeomorphism invariance (and, more generally, in terms of gauge symmetries). One difficulty with naively trying to quantize gravity at the Archimedean place is that the gravitational degrees of freedom are mixed by diffeomorphism invariance. Attempts to isolate the physical degrees of freedom, and then to quantize them, have so far not been successful. This difficulty is of the same type as isolating the degrees of freedom in gauge theories, by fixing, or otherwise controlling, the gauge symmetries. However, while for gauge theories techniques for dealing with the gauge do exist, for gravitational theories the question seems considerably harder.

Performing a $p$-adic decomposition of gravity effectively removes diffeomorphism invariance from the problem. This is because if the fundamental system is $p$-adic, diffeomorphism invariance is not there to begin with.[18] Rather, diffeomorphism invariance is only an emergent phenomenon that appears when all places are put together. For Archimedean two-dimensional and three-dimensional hyperbolic spaces, $p$-adic decomposition thus translates into working with the group $\mathrm{SL}_2(\mathbb{Q}_p)$ (and its direct product with itself) over finite places, rather than with the group $\mathrm{SL}_2(\mathbb{R})$. We will come back to a technical analysis of two dimensional gravity in Section 4.[19]

It is natural to ask whether gravity as a $p$-adic theory can be quantized on its own, without embedding it into a larger theory such as string theory. I will not have an answer to this question in the present paper.

Let's now discuss the connection to string theory. Supersymmetry is neither apparent nor required at the basic level of discussion in this paper, however it is known that Archimedean string theory contains gravity (and, indeed, Archimedean string theory can be thought of as an attempt at a UV completion of gravity). It is thus straightforward to conjecture that, as a $p$-adic theory, gravity should still emerge from string theory, and thus that string theory should admit a $p$-adic decomposition. Furthermore, given the AdS/CFT correspondence, as well as the unnaturalness of separating theories

---

[18]Note, however, that the tree does have an automorphism symmetry that can be thought of as a kind of "residual" symmetry of diffeomorphism invariance.

[19]A natural question is whether $\mathrm{SL}_2(\mathbb{Q}_p)$ and its associated Bruhat-Tits building are sufficient, or whether on the $p$-adic side we should expect some sort of enhancement to a bigger symmetry group. This question is related to whether the role of the bulk can played by other objects, such as Drinfeld spaces.

into gravitational and non-gravitational beyond certain limits, it also seems reasonable to conjecture that generic quantum field theories should admit $p$-adic decompositions.

There is a bit of lore, related to the discussion above, that I would like to comment on. It is often said that the Archimedean AdS/CFT correspondence shows that black hole evaporation is unitary, since it is dual to a field theory process. This slogan is misleading, in the following sense. Quantum field theory evolution should indeed be unitary, if the field theory is a mathematically well-defined theory. It is not. It can (and should!) be argued that unitarity is present if the field theory is properly UV-completed with the heavy operators and other objects, however the question then becomes what this UV-completion should be, and whether it could be non-Archimedean. In this sense, the $p$-adic proposal of this paper is thus compatible with what is already known from AdS/CFT.

As a philosophical point, what the AdS/CFT correspondence shows in this context is that the field theoretic side is not understood, and that the difficulties in rigorously defining quantum field theories mathematically are related to the difficulties in understanding the degrees of freedom in quantum gravity.

Of course, it should not be expected that all theories can only be UV-completed $p$-adically. For instance, it is known that topological field theories can be defined rigorously on the Archimedean side, and the free particle example of Section 2 shows that, at least in simple cases, it is possible to have both $p$-adic and Archimedean descriptions of the same theory.

Stepping back to general theories, how should the connection between the $p$-adic and Archimedean worlds be realized? The rule is that for every instance of $p$-adic decomposition, there should exist mathematical identities relating the quantities at the Archimedean and finite places, possibly also with some adelic contributions. The validity of these identities should of course be independent of the physical content of the theories. Euler products are a simple illustration of such identities, but in general things could be more complicated. A recent example in this direction has been worked out by references [59,60], in the context of Zeta functions and of explaining the formal $p \to 1$ limit of $p$-adic string theory, for string Lagrangians and amplitudes (for more details on the $p \to 1$ limit in string theory see e.g. [61]).

Finally, let me also remark that my proposal in this paper is essentially a *framework*. So far I have mostly been talking about $\mathbb{Q}_p$ (and implicitly about the Bruhat-Tits tree associated to $\mathrm{SL}_2(\mathbb{Q}_p)$), however there are other non-Archimedean objects that can be

used. A natural generalization of the tree is the Drinfeld upper half-plane, which can be thought of as the usual upper half-plane with a nontrivial topology, or, colloquially, as the upper half plane with "a hole at every point;" for a rigorous introduction see e.g. [62]. Furthermore, it is known that prime ideals are needed to describe Virasoro-Shapiro amplitudes [14].

## 3.2 More dictionary entries

I will now comment on some more dictionary entries between the non-Archimedean and Archimedean worlds.

**Classical configurations and quantum states:** As already mentioned in Section 2, for an Archimedean theory that $p$-adically decomposes, the least we can demand is that for any classical configuration there should exist $p$-adic configurations at some finite places from which the Archimedean configuration can be reconstructed. Furthermore, if the Archimedean theory has a notion of time evolution, it should be reconstructible from time evolution at the finite places, and the time evolved $p$-adic configurations should map to the time-evolved Archimedean configuration. If the $p$-adic theory is quantum, the classical configuration should be reconstructible from $p$-states, and if the Archimedean theory is quantum, then the Archimedean states should be reconstructible from $p$-states. Conversely, it is not immediate that any $p$-states over some finite places correspond to an Archimedean state, or to a classical configuration. It would be interesting to understand the physical interpretation of such general $p$-states, and in particular, it would be interesting to explore whether Archimedean states can be regarded as some kind of diagonal embedding among general $p$-states.

**Sparseness of $p$-adic reconstruction**: In the free particle example of Section 2, and in the Euclidean two-dimensional gravity of Section 4 below, the decomposition over the finite places is relatively simple, in that the physics at all the places is the same.[20] This is not a general feature. The theories at the finite places can exhibit different behavior, depending on the prime. This was already apparent in [50], where the symmetry group and commutation relations change with the prime and sign function being used, and will also be the case for the Lorentzian two-dimensional gravity of Section 5 below.

---

[20]This statement is almost true: place $p = 2$ for the free particle is special, in that formulas must be rederived, but end up morally the same as at the higher $p$ places, and there are also some slight special features at places $p = 2, 3$ in the gravitational story.

A natural question thus is whether Archimedean physics must be reconstructed from all places, or whether certain places suffice (such as the places with the correct symmetry group and commutation relations in the context of [50]). While I won't have a definitive answer in this paper, it is illustrative to discuss a simple example of how reconstruction from only certain places could work. Suppose we are attempting to reconstruct an Archimedean function out of a set of places $\{p_i\}$. If the set of rationals that have all prime factors in $\{p_i\}$ is dense in $\mathbb{Q}$, then the function at the Archimedean place can be reconstructed over all $\mathbb{R}$.[21] This can be seen straightforwardly e.g. in the case of the norm, however it will also work when the function depends on digits in the $p$-adic expansion. Of course, there has to be an a priori reason for why the reconstruction should work, i.e. for why the function recovered at the Archimedean place should be a continuous function with the desired properties (or, alternatively, this procedure can be used to test for $p$-adic reconstruction).

What is the minimum number of primes that can be used? It may seem that at least an infinite number of primes is needed to obtain a set of rationals dense in $\mathbb{Q}$, however it turns out that two primes suffice, in that the set of rationals with prime factors $p_{1,2}$, for any two fixed primes, is dense in $\mathbb{Q}$.[22] Furthermore, if we allow analytic continuation in the valuation, then in some cases it is possible (and useful) to take the $p \to 1$ limit (see e.g. [59, 60] for recent work in this direction).

This sparseness property, that few places can reconstruct physics over the entire Archimedean domain, may appear strange, however it has an important significance: complicated number theoretic objects often have interesting behavior only over a small (finite) number of places, and are trivial at an infinite number of places.[23] The sparseness of $p$-adic reconstruction thus enhances the set of Archimedean theories that can be reconstructed $p$-adically, in that it allows Archimedean physics to take full advantage of complicated number theoretic objects, regardless of over how few places they may exhibit nontrivial behavior. This is true whether some, or all, places are used in the reconstruction. Similarly, if an adelic cutoff ends up being required above which all physics is trivial, the sparseness property ensures that the Archimedean physics won't be affected at the level of the discussion in this section. Note, however, that the cutoff could still appear on the Archimedean side; we will come back to this point in Section 6.

---

[21]Just as in Section 2, we are not attempting reconstruction at any Archimedean irrationals.

[22]A proof of this fact follows from Kronecker's theorem, as explained by [63] (see Theorem 2.1).

[23]The the sign functions of Section 2 also exhibit this behavior.

From the discussions in Sections 2, 5, and in [50], the finite places with physics most similar to the Archimedean physics are for primes $p = 3 \mod 4$ and parameter $\tau$ corresponding to totally ramified quadratic extensions. However, it is not clear if these places are sufficient (or even required) to reconstruct Archimedean physics generally.

**Mapping Hamiltonians and Lagrangians:** For quantum mechanical and field theories, it is possible to give a tentative prescription for how the Lagrangians and Hamiltonians map when passing from Archimedean to $p$-adic, at least for simple systems. The prescription can be read off from Section 2: replace the ordinary derivatives with Vladimirov derivatives, and keep the polynomial potential term, placing norms on all non-derivative quantities that are not target-space valued. Schematically,

$$\partial_x^s, \; \partial_t \quad \leftrightarrow \quad \partial_x^{s,\tau_x}, \; \partial_t^{1,\tau_t} \tag{3.1}$$
$$m, \; x \quad \leftrightarrow \quad |m|, \; |x|$$

with the left-hand side Archimedean and the right-hand side $p$-adic. Here $\tau_x$ and $\tau_t$ parameterize the sign characters with which the derivatives are twisted. For the free particle discussion in Section 2 we had $\tau_x = 1$ and $\tau_\tau = -1$; this choice appears to be dictated by recovering the Lorentzian signature on the Archimedean side, so it will be interesting to investigate if it persists for other quantum mechanical Hamiltonians also. This prescription of replacing regular derivatives with Vladimirov ones has also been employed e.g. in [50].

For gravity the rules seem to be different, as there are no Vladimirov derivatives on the bulk side. At least in the Euclidean case (which will be discussed in Section 4), the rules which seem to be universal across places are the prescription for obtaining Ricci curvature from the Wasserstein distance, and the curvature entering the Lagrangian.

**Archimedean equations of motion:** In Section 2, the equation of motion obtained by $p$-adic reconstruction exactly matched the Archimedean Schrödinger equation. This is a likely general feature, in that there should exist a regime in which $p$-adic reconstruction mostly recovers the Archimedean equations of motion, via an Ehrenfest-like theorem. However, the underlying dynamics are $p$-adic, and this dynamics could be very different from Archimedean dynamics in other regimes. Understanding precisely when, and how, the Archimedean equations of motion are recovered is an important and nontrivial question, but I will not address it in the rest of the paper.

**Bruhat-Tits buildings:** One possible generalization of the tree gravity to higher

dimensions is given by Bruhat-Tits buildings. Buildings are simplicial complexes, often associated to certain groups, which for our purposes can be thought of as generalizations of Riemannian symmetric spaces to non-Archimedean settings, i.e. the spaces to which the gravitational theory should apply. The natural proposal then is to identify the symmetry group across places, just as in the tree/AdS$_2$ case SL$_2$ was acting at all places.[24] This identification of symmetry groups should apply to the vacuum geometries. Given a theory of gravity, the buildings should be dynamical, and allowed to deform away from the vacuum configuration.[25] More details on buildings can be found in [64].

Other generalizations are also likely to exist, such as the one obtained by replacing buildings with Drinfeld symmetric spaces, which in some intuitive sense is the higher dimensional analogue of passing from the Bruhat-Tits tree to the $p$-adic upper half-plane. For a discussion of certain Drinfeld symmetric spaces see e.g. [65].

**Field theory:** The non-Archimedean field theories that reconstruct Archimedean field theories in one dimension are defined on $\mathbb{Q}_p$. In higher dimensions, if gravity is reconstructed from Bruhat-Tits buildings, then the field theories should also be defined on the buildings. In the context of AdS/CFT, gravity on buildings is thus dual to a CFT living on the boundary, but it is also possible to define field theories in the bulk of the buildings.

It is important to emphasize that other $p$-adic constructions of field theories also exist. For instance, an axiomatic approach to constructing non-Archimedean scalar fields on a $p$-adic analogue of Minkowski space was recently considered by [66].

**Lorentzian signature:** In the context of the free particle in Section 2 and of Section 5 below, Archimedean Lorentzian signature seems to be associated with quadratic extensions of $\mathbb{Q}_p$. This translates into a kind of "dressing" of the edges of the Bruhat-Tits tree, according to signature. However, as already pointed out in [50], the quadratic extension parameter also affects the types of theories allowed at the finite places.

---

[24]The reader should beware that there is also a modding by a maximal compact subgroup present here.

[25]This will require generalization away from the strict definition of a building.

# 4  Euclidean $\mathrm{AdS}_2$ from finite places

In this section and the next I will consider how the $p$-adic framework applies to Euclidean and Lorentzian gravity. Although understanding the connection between Archimedean gravitational microstates and the $p$-adic decomposition of gravity is an important question, in these two sections I will only consider semiclassical gravity.

The Euclidean proposal is that the Bruhat-Tits tree $T_{\mathrm{SL}_2(\mathbb{Q}_p)}$ is the $p$-adic analogue of $\mathrm{EAdS}_2$,[26] and that a genus $g$ configuration on the Archimedean side can be reconstructed from genus $g$ configurations at the finite places. The Lorentzian proposal (in Section 5) is that the tree with symmetry group $\mathrm{SL}_2(\mathbb{Q}_p[\sqrt{\tau}])$ acting on the boundary on a quadratic extension $\mathbb{Q}_p[\sqrt{\tau}]$ of the $p$-adics is the analogue of Lorentzian $\mathrm{AdS}_2$, with a distinguished Poincaré wedge in the Archimedean geometry being analogous to the embedding of $T_{\mathrm{SL}_2(\mathbb{Q}_p)}$ inside $T_{\mathrm{SL}_2(\mathbb{Q}_p[\sqrt{\tau}])}$.

## 4.1  Review

Let's discuss the Euclidean case. The details of gravity on $T_{\mathrm{SL}_2(\mathbb{Q}_p)}$ and other graphs have been worked out[27] in [45] (for a review of Bruhat-Tits trees see e.g. [67]). Here is a rapid review of that story. Vertices in the graph are denoted by lowercase Latin letters $x$, edges $\langle xy \rangle$ by angled brackets (this notation assumes that $x$ and $y$ are neighbors in the graph), and $x$ and $y$ being neighbors is denoted by $x \sim y$. The curvature on edge $\langle xy \rangle$ is denoted by $\kappa_{\langle xy \rangle}$, and it is the analogue of Archimedean Ricci curvature. It is useful to introduce the objects

$$J_{\langle xy \rangle} := \frac{1}{a_{\langle xy \rangle}^2}, \quad d_x := \sum_{y \sim x} J_{\langle xy \rangle}, \quad c_x := \sum_{y \sim x} \sqrt{J_{\langle xy \rangle}}. \qquad (4.1)$$

The fundamental degree of freedom is the edge length $a_{\langle xy \rangle} > 0$ of any edge $\langle xy \rangle$ in the Bruhat-Tits tree. By introducing a Wasserstein distance $W(\psi_1, \psi_2)$ between two probability distributions $\psi_1(x)$, $\psi_2(x)$, defined on the vertices of the graph and demanding that, in the limit where the probability distributions are sharply peaked around two neighboring points, the curvature enters the Wasserstein distance in the

---

[26]This was already recognized by [49].

[27]As explained in [45], this construction is *one* way of introducing gravity on trees, and it is conceivable that other ways of defining gravity could exist, either for Einstein gravity or for some of its cousins, such as topological massive gravity or Jackiw-Teitelboim gravity.

same way as in the Archimedean case, the graph curvature can be read off as

$$\kappa_{\langle xy\rangle} = \frac{1}{d_x a_{\langle xy\rangle}}\left(\frac{1}{a_{\langle xy\rangle}} - \sum_{\substack{z\sim x\\z\neq y}}\frac{1}{a_{\langle xz\rangle}}\right) + \frac{1}{d_y a_{\langle xy\rangle}}\left(\frac{1}{a_{\langle xy\rangle}} - \sum_{\substack{z\sim y\\z\neq x}}\frac{1}{a_{\langle yz\rangle}}\right). \qquad (4.2)$$

Formula (4.2) is valid for uniform valence graphs that have no loops, and for graph with loops provided that the change in lengths $a_{\langle xy\rangle}$ along each loop is small enough relative to the number of vertices on the loop. The details of this limitation are spelled out in [45], and will not be important for the purposes of this section.

Denote a region of the graph by $\Sigma$, such that its boundary $\partial\Sigma$ consists only of vertices. The gravitational action for this region is

$$S_\Sigma = \frac{1}{16G_N^{(p)}}\left(\sum_{\langle xy\rangle\in\Sigma}\kappa_{\langle xy\rangle} + \sum_{x\in\partial\Sigma}k_x\right), \qquad (4.3)$$

where $k_x$ is a boundary term integrand (analogous, up to a factor of 2, to the extrinsic curvature in the Archimedean case), and $G_N^{(p)}$ is a $p$-adic Newton's constant. As usual, the gravitational equations of motion follow from requiring that the action is stationary under arbitrary variations of the edge lengths inside $\Sigma$, and the boundary extrinsic curvature is determined by demanding stationarity of the action (and the same equations of motion) for the edges neighboring boundary $\partial\Sigma$.

## 4.2 Matching partition functions

Since our analysis is semiclassical, from the general discussion in Section 3, we expect that the matching of Euclidean partition functions should be performed around saddles, with the Archimedean saddle reconstructible from the $p$-adic ones. We will only consider $p$-adic saddles for uniform edge lengths; these are constant negative curvature solutions to the Einstein equations, as explained in [45]. The question of determining all $p$-adic saddles allowed by the $p$-adic Einstein equations is left for future work.

Consider a configuration of the uniform valence graph that has all edge lengths constant, and genus $g$. At least in certain cases, it is possible to think of such a configuration as arising from quotienting $T_{\mathrm{SL}_2(\mathbb{Q}_p)}$ by a $p$-adic Fuchsian-Schottky group, but this will not be required at the level of the discussion in this section. If the genus

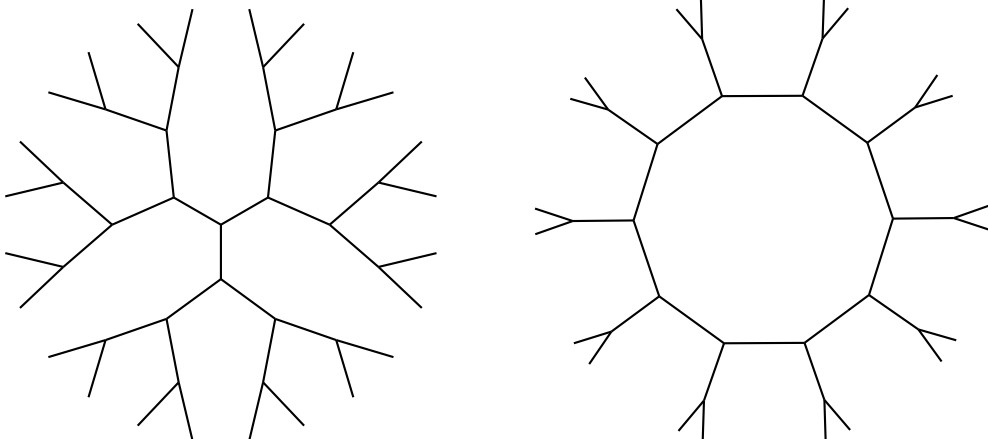

Figure 1: Left: The Bruhat-Tits tree $T_{\mathrm{SL}_2(\mathbb{Q}_p)}$ is an infinite graph of uniform valence $p + 1$ ($p = 2$ on the figure), with symmetry group $\mathrm{SL}_2(\mathbb{Q}_p)/\mathrm{SL}_2(\mathbb{Z}_p)$ and boundary $P^1(\mathbb{Q}_p)$. Right: The BTZ graph is obtained by quotienting the tree with a Schottky group, which maps one of the tree's geodesics into a loop. For more details see e.g. [44].

is trivial, such a configuration is just the Bruhat-Tits tree $T_{\mathrm{SL}_2(\mathbb{Q}_p)}$, and the genus $g = 1$ graph can be thought of as an analogue of the BTZ black hole (see Figure 1), although the Archimedean manifolds being reconstructed will be two-dimensional, as we now explain.

For any genus $g$ configuration of uniform edge lengths, the action (4.3) is topological, in that it evaluates to (see [68] for the details),

$$S^{(p)}(g) = -\frac{g - 1}{8G_N^{(p)}}. \tag{4.4}$$

Result (4.4) is analogous to the Gauss-Bonnet theorem in two Euclidean dimensions. More specifically, consider the realization of EAdS$_2$ as a Poinaré disk $\mathcal{P}$, and include $g$ punctures of finite size (see Figure 2). Then the Gauss-Bonnet theorem states that the action

$$S^{(a)} = \frac{1}{16\pi G_N^{(a)}} \int_{\mathcal{P}} R + \frac{1}{8\pi G_N^{(a)}} \int_{\partial \mathcal{P}} K \tag{4.5}$$

is topological, evaluating to

$$S^{(a)}(g) = -\frac{g - 1}{8G_N^{(a)}}. \tag{4.6}$$

Thus, provided that the $p$-adic and Archimedean Newton's constants satisfy the relation

$$\frac{1}{G_N^{(a)}} = -\sum_p \frac{1}{G_N^{(p)}},$$
(4.7)

the actions sum to zero across places and the adelic identity

$$Z^{(a)}(g) = \prod_p \frac{1}{Z^{(p)}(g)}$$
(4.8)

holds for every genus $g$.

The interpretation of Eq. (4.8) is very simple: the finite places act as independent physical systems, out of which the Archimedean system arises.

A few comments are now in order. First, a cosmological constant has not been included in Eq. (4.3), in analogy with Archimedean 2d gravity. It is in fact possible to include such a term, but it does not change the qualitative features of the discussion, although it will enter some of the equations, such as the matching (4.7). At finite places, it seems to be the case that the nature of the tree itself is responsible for the negative curvature; the cosmological constant term in the action cannot have this role, since it does not enter the equations of motion. Second, the boundary term in the action behaves differently at the Archimedean and finite places. In the Archimedean case, the boundaries of the punctures contribute the extrinsic curvature to the action, whereas at a finite place the vertices on any loop have the same valence as vertices on the branches that extend to infinity, are thus part of the bulk, and do not provide boundary term contributions.

## 4.3  Reconstructing the Archimedean Einstein equations and geometry

The Einstein equations arising from action (4.3) are nontrivial and independent of the value of the cosmological constant. This is in stark contrast with the Archimedean case, where in two dimensions for pure gravity the Einstein equations are trivial without a cosmological constant, and incompatible with one.

How to interpret this discrepancy? It is useful to remember the free particle example of Section 2, where the $p$-adic time evolution gave rise to *almost* the usual Archimedean time evolution, in that time evolution was recovered with the additional

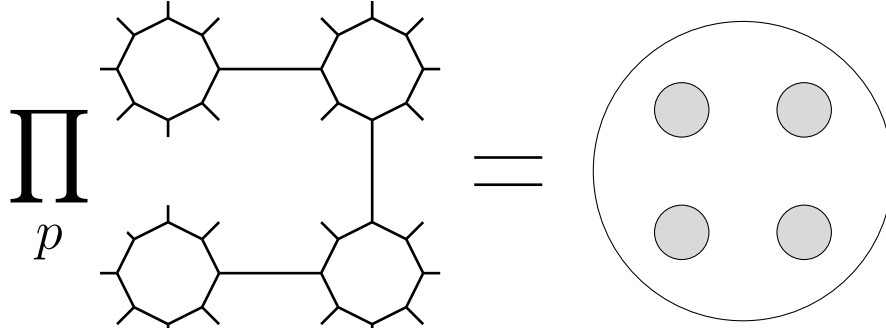

Figure 2: Cartoon of how Euclidean $AdS_2$ with $g$ punctures arises from the trees at finite places trees. The product $\prod_p$ refers to the fact that the Archimedean partition function is the inverse product of the finite place partition functions. The reader should keep in mind that this is a cartoon only, and the precise way in which the reconstruction happens will be more complicated (see discussion at the end of Section 4.3).

constraint $m > 0$. Similarly, in the case of Euclidean two-dimensional gravity, the finite places give rise to *almost* Archimedean Einstein gravity, in the sense that the resulting Archimedean Einstein equations are trivial, but not all manifolds are allowed. Rather, the only allowed manifolds are those that can arise from $p$-adic configurations obeying the $p$-adic Einstein equations.[28] I will not determine in this paper precisely which manifolds these are, or how the $p$-adic reconstruction should be performed for arbitrary genus. However, it is possible to outline how the reconstruction should function, in the case of the tree.

It is often said that pure two-dimensional gravity as an Archimedean theory is not well-defined, because the Einstein equations are vacuous. The analysis in this section suggests, however, that pure two-dimensional Archimedean gravity, as a theory reconstructed from the finite places, is well-defined.

Let's now turn to saddle reconstruction. The ansatz is that Euclidean vacuum $AdS_2$ $p$-adically decomposes into $T_{\mathrm{SL}_2(\mathbb{Q}_p)}$ trees. The $AdS_2$ can then be reconstructed from the finite places, by considering the geodesics between two boundary points $x, y \in \mathbb{Q}$ (just as in the discussions in Sections 2 and 3, on the boundary we restrict to points that are common to all $\mathbb{Q}_p$'s, that is to points in $\mathbb{Q}$). The geodesic length in the tree (see e.g. [44]) is

$$\ell_p(x,y) = 2\log_p \frac{|x-y|_p}{\epsilon_p},\tag{4.9}$$

---

[28]It is natural to conjecture that in higher dimensions the $p$-adic Einstein equations will match directly to the Archimedean ones.

with $\epsilon_p$ a cutoff. Abstracting of this cutoff, the geodesic length $\ell_p(x, y)$ thus determines the $p$-adic norm $|x - y|_p$. Putting all places together determines the Archimedean norm $|x - y|_\infty$, from which the Archimedean geodesic length can be determined via the formula

$$\ell_\infty(x, y) = 2 \ln \frac{|x - y|}{\epsilon}, \tag{4.10}$$

where an Archimedean cutoff $\epsilon$ has been introduced. Formula (4.10) holds for rational boundary points, however by continuity it can be extended to irrationals points also.

The trees at all finite places thus determine the geodesics between any two boundary points at the Archimedean place. But in the case case of two dimensional Riemannian manifolds, knowledge of all geodesics is enough to reconstruct the manifold, without any assumption on curvature [69]. In this sense, the $p$-adic $T_{\mathrm{SL}_2(\mathbb{Q}_p)}$ saddles at the finite places uniquely determine the Euclidean vacuum AdS$_2$ at the Archimedean place.

Moving up in genus, for BTZ graphs the moduli of the central ring at the finite places should determine the Archimedean modulus of the puncture, although I will not work out in this paper how this happens. For higher genus configurations, the moduli of the cycles should similarly determine the moduli of the allowed Archimedean saddles.

Understanding how the moduli get mapped to the Archimedean side is important for understanding the precise Archimedean saddles that can arise. The value of the saddle (as in Eq. (4.8)) by itself does not in general uniquely specify the Archimedean manifold.[29] That is, although in the discussion above I was matching genus $g$ graphs to Euclidean $AdS_2$ with $g$ punctures, some other Archimedean manifolds, with the same values of the action, could end up the correct Archimedean spaces being reconstructed. In particular, the correct objects may be genus $g$ Riemann surfaces, with certain choices of metric. At the level of only looking at the partition function value, different manifolds with the same on-shell action are generally not distinguishable.

## 5   Lorentzian AdS$_2$ from quadratic extensions

In this section I would like to propose a Lorentzian version of the Bruhat-Tits tree, together with curvature, action and edge equations of motion. The natural object to consider for this proposal is the Bruhat-Tits tree for $\mathrm{SL}_2(\mathbb{Q}_p[\sqrt{\tau}])$, with $\mathbb{Q}_p[\sqrt{\tau}]$ a

---

[29]It is worthwhile to remark that reconstruction to the Archimedean side may not be possible for all values of the $p$-adic moduli. Rather, the graphs arising from quotienting by $p$-adic Fuchsian-Schottky groups could play a privileged role.

quadratic extension of $\mathbb{Q}_p$.

On the Archimedean side, the object we will be interested in is not quite $\mathrm{AdS}_2$, but rather $\mathrm{AdS}_2$ with a distinguished Poincaré wedge. This is because the tree for $\mathrm{SL}_2(\mathbb{Q}_p)$ sitting inside $\mathrm{SL}_2(\mathbb{Q}_p[\sqrt{\tau}])$ is similar to a Poincaré wedge sitting inside $\mathrm{AdS}_2$, so we will simply identify the two as such, for the purposes of Archimedean reconstruction. There is a certain sense in which specifying the wedge does not commute with the global $\mathrm{SL}_2$ symmetry. This is already true on the Archimedean side, but seems to become even more pronounced at a finite place. This is a rather strange feature, and is likely indicative of the fact that diffeomorphism invariance arises on the Archimedean side after the $p$-adic reconstruction is performed, but we will not pursue it further in this paper.

Let's get down to the details. What we would like to obtain is a $p$-adic decomposition of Lorentzian $\mathrm{AdS}_2$ (with a distinguished wedge), however an immediate difficulty is that it is not obvious how the finite places should be put together.[30] For this reason, we will restrict our analysis to individual finite places, and we will take as guidelines that the quadratic extension trees are the correct objects to reconstruct Lorentzian $\mathrm{AdS}_2$ the following features:

1. There will be two types of edges, or *three* if we also count certain differences in the equations of motion.

2. There is a qualitative analogy between certain objects defined on $\mathrm{AdS}_2$ with a distinguished wedge, and objects defined on the quadratic extension tree. This will be discussed in Section 5.6.

3. The operator entering the linearized Einstein equations for the graviton is balanced, in the same way the tree Laplacian in [45] is balanced. This will be interpreted as an indication that the graviton is massless.

4. A certain sign in the operator entering the linearized Einstein equations flips, depending on whether the edge is in $T_{\mathrm{SL}_2(\mathbb{Q}_p)}$, or in $T_{\mathrm{SL}_2(\mathbb{Q}_p[\sqrt{\tau}])} - T_{\mathrm{SL}_2(\mathbb{Q}_p)}$. We will take this as a hint that the operator switches between being elliptic and hyperbolic.

---

[30]Meaning that it is not clear what the value of $\tau$ should be, which determines how the unramified and totally ramified places are distributed. It is possible that for the purpose of reconstructing Archimedean physics the value of $\tau$ is simple to pick (such as deciding between $\tau > 0$ and $\tau < 0$ at the Archimedean place), but this will not be investigated here.

It is important to emphasize that while these features are *suggestive*, they don't go all the way in establishing the quadratic extension trees as the correct objects from which AdS$_2$ can be reconstructed. In particular, certain elements of the trees differ from those of Archimedean AdS$_2$. Let's remark on the following:

1. It is not clear how the causal structure of AdS$_2$ arises from the tree. This is related to understanding how Lorentzian correlators can be reconstructed from the trees. A connected technical point is that although I will use below the terminology of *spacelike*, *timelike*, and *horizon* edges on the tree, this does not imply that tree correlators computed over such separations will match their Archimedean counterparts, nor does it imply that they *should*. Rather, the rules of the game are that Archimedean correlators should be reconstructible from their *p*-adic counterparts. Understanding precisely how this happens will be left for future work.

2. Although I denote the second order operators on the tree arising in the linearized Einstein equations as *elliptic* and *hyperbolic*, I will not explore in this paper precisely how similar they are to Archimedean elliptic and hyperbolic operators. However, in order for AdS$_2$ to indeed be reconstructible from the trees, it should be expected that many of the properties of Archimedean hyperbolic and elliptic operators should continue to hold.

3. The trees exhibit a certain collapse of sectors: at almost all places, all edges inside the wedge are timelike, and all edges outside are spacelike. This is in stark contrast with AdS$_2$, where every point in the bulk has timelike, null, and spacelike directions tangent to it.

These issues, and more, will need to be understood in order to make sense of how, and if, the trees reconstruct AdS$_2$.

An immediate question is why we are singling out quadratic extensions for the reconstruction of wedges inside AdS$_2$, out of all possible *n*-th order extensions. In fact, it is natural to ask whether arbitrary extensions could be useful for reconstructing more complicated wedge configurations inside AdS$_2$, but this is also a direction I will not pursue further.

## 5.1   $\mathrm{SL}_2(\mathbb{Q}_p[\sqrt{\tau}])$ trees

Since $\mathbb{Q}_p[\sqrt{\tau}]$ is a quadratic extension, it is either unramified or totally ramified (for a review see e.g. Appendix C). The Bruhat-Tits tree in this case is an infinite tree of uniform valence $Q + 1$, with $Q = p^2$ in the unramified case, and $Q = p$ in the ramified case. There exists a natural embedding of the Bruhat-Tits tree for $\mathrm{SL}_2(\mathbb{Q}_p)$ inside the tree for $\mathrm{SL}_2(\mathbb{Q}_p[\sqrt{\tau}])$, as shown in Figure 3, with the solid edges in $T_{\mathrm{SL}_2(\mathbb{Q}_p)}$, and the dashed edges in $T_{\mathrm{SL}_2(\mathbb{Q}_p[\sqrt{\tau}])} - T_{\mathrm{SL}_2(\mathbb{Q}_p)}$. We can define a solid (or dashed) geodesic as any path between $x, y \in \partial T_{\mathrm{SL}_2(\mathbb{Q}_p[\sqrt{\tau}])}$ that only travels along solid (or dashed) edges. In the unramified case, each vertex on a solid geodesic connects to $p + 1$ solid edges, and to $p^2 - p$ dashed edges. In the ramified case, vertices along a solid geodesic alternatingly connect to 2 solid edges and $p - 1$ dashed edges, or to $p + 1$ solid edges and no dashed edges. Vertices along a dashed geodesic always connect to $Q + 1$ dashed edges.

What should the Lorentzian structure be in the bulk? A natural guess is that we now have two types of edges neighboring each vertex $x$: $q_x^+$ *spacelike* edges $\langle xy \rangle$ with length squared $a_{\langle xy \rangle}^2 > 0$, and $q_x^-$ *timelike* edges $\langle xy \rangle$ with length squared $a_{\langle xy \rangle}^2 < 0$, with

$$q_x^+ + q_x^- = Q + 1. \tag{5.1}$$

We must establish how these edges are distributed in the $\mathrm{SL}_2(\mathbb{Q}_p[\sqrt{\tau}])$ tree. It may seem desirable to demand that $q_x^+$ and $q_x^-$ are uniform at all vertices on the tree, however there doesn't seem to exist a natural action of $\mathrm{SL}_2$ on the tree that preserves the types of edges. Said another way, under a generic element of $\mathrm{SL}_2$ an edge $\langle xy \rangle$ with $a_{\langle xy \rangle}^2 > 0$ can get mapped into one with $a_{\langle xy \rangle}^2 < 0$. This is in contrast with Archimedean Lorentzian $\mathrm{AdS}_2$, where the conformal group preserves the interior and exterior of the lightcone of a point in the bulk.

So what we should do instead is follow the symmetry. In both the unramified and totally ramified cases, the sets of dashed and solid edges are preserved by the action of $\mathrm{SL}_2(\mathbb{Q}_p)$ on $T_{\mathrm{SL}_2(\mathbb{Q}_p[\sqrt{\tau}])}$, since the solid subtree terminates on points in $P^1(\mathbb{Q}_p)$ (i.e. points that don't contain $\sqrt{\tau}$), and the action of the group doesn't introduce any $\sqrt{\tau}$ terms. Let us *define* the solid edges as timelike, and the dashed as spacelike. A timelike geodesic is any path in the tree between $x, y \in \partial T_{\mathrm{SL}_2(\mathbb{Q}_p[\sqrt{\tau}])}$ that only travels along timelike edges, and similarly for spacelike geodesics (timelike and spacelike geodesics are thus the same as solid and dashed geodesics, respectively). We can think of moving along solid geodesics as translating in time a copy of the dashed spacelike branches.

It is possible to make a further distinction, and count the spacelike edges that neighbor timelike edges separately. Then in both the unramified and totally ramified cases, we have three types of edges in the Lorentzian tree:

1. Timelike edges, e.g. $\langle zw \rangle$ in Figure 3, obeying $a^2_{\langle zw \rangle} < 0$.

2. Spacelike edges, e.g. $\langle xy \rangle$ in Figure 3. These edges do not neighbor timelike edges, and have $a^2_{\langle xy \rangle} > 0$.

3. Horizon edges, e.g. $\langle su \rangle$ in Figure 3. These have spacelike signature ($a^2_{xy} > 0$) and neighbor both timelike and spacelike edges. We choose to count them separately because the equations of motion for them will exhibit certain features not present for the spacelike edges. The term *horizon* is being used loosely in this context, motivated by the fact that these edges connect between the inside and outside of the wedge; there are certain features of Archimedean horizons which horizon edges do not share. In particular, while it may seem natural to define a third length scale for horizon edges, for trees of uniform length we will take the horizon edge lengths to equal the lengths of the spacelike edges. The reason for this is that defining a separate length scale for the horizon edges will not be compatible with the Einstein equations. Furthermore, the linearized Einstein equations will couple the horizon and spacelike edges.

From now on referring to the spacelike edges will not include the horizon edges, and referring to spacelike signature edges will include both the spacelike and horizon edges.

The structure of the tree is very different from the usual Archimedean Lorentzian AdS$_2$. For each spacelike branch, there is a distinguished root vertex that connects to the timelike geodesic, and all other vertices have no timelike geodesics passing through.

## 5.2   Defining curvature and action

We would like to define and compute curvatures, in the spirit of [45]. It no longer makes sense for the starting point to be a Wasserstein distance, since in the Lorentzian setting two remote points can have zero or arbitrary negative separation. Rather, since we already have a notion of curvature (4.2) in the Euclidean setting, we can start from there and try some adjusting. Let's propose some minimal modifications. Since in [45] $d_x$ was defined as a neighbor sum of inverse $a_{\langle xy \rangle}$ squared, it makes sense to preserve

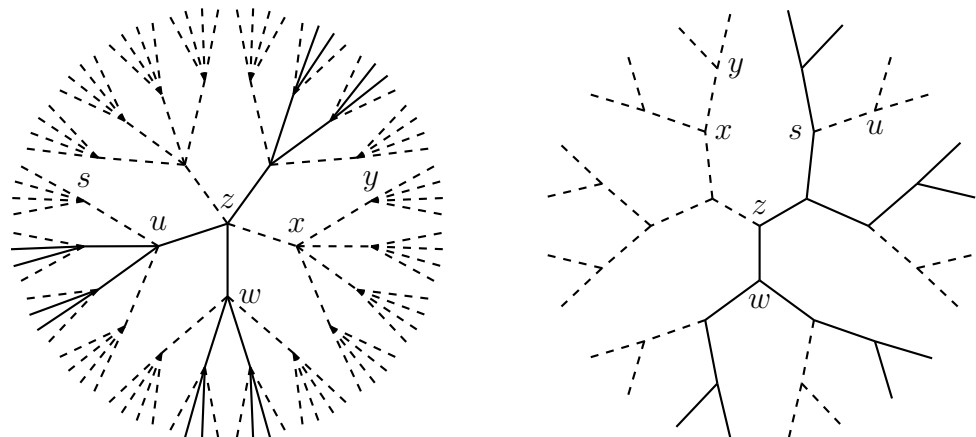

Figure 3: The unramified and totally ramified trees for $\mathrm{SL}_2(\mathbb{Q}_p[\sqrt{\tau}])$. The valence of the unramified tree is $p^2+1$, and that of the ramified tree is $p+1$ ($p=2$ on the figure). The embedding of $T_{\mathrm{SL}_2(\mathbb{Q}_p)}$ inside $T_{\mathrm{SL}_2(\mathbb{Q}_p[\sqrt{\tau}])}$ is denoted by solid edges, with the dashed edges belonging to $T_{\mathrm{SL}_2(\mathbb{Q}_p[\sqrt{\tau}])} - T_{\mathrm{SL}_2(\mathbb{Q}_p)}$. We identify the solid edges as timelike and the dashed edges as spacelike.

this definition, only now we will sum over the negative edge squares also. Thus, we declare

$$d_x = \sum_{y \sim x, \pm} \frac{1}{a_{\langle xy \rangle}^2}, \tag{5.2}$$

where the $\pm$ is an instruction to sum over edges of all signature.

In order to define curvature we need to introduce the following barred index notation. This notation is only used when there exists a preferred edge $\langle xy \rangle$. Then terms with an even number of bars have the same signature as $a_{\langle xy \rangle}^2$, and the terms with odd number of bars have opposite signature, as in Figure 4. Similarly, $q_x$ is the number of edges of the same signature as edge $\langle xy \rangle$ at vertex $x$, and $\bar{q}_x$ the number of edges of opposite signature. If we refer to the number of edges of a particular signature, we employ $q_x^{\pm}$ instead.

Let's now define curvature and the gravitational action. For the curvature we propose the expression

$$
\begin{aligned}
\kappa_{\langle xy \rangle}^L &= -\frac{1}{d_x} \left( \frac{1}{a_{\langle xy \rangle}^2} + \sum_{k=1}^{q_x-1} \frac{1}{a_{\langle xy \rangle} a_{\langle xx_k \rangle}} + \sum_{k=1}^{\bar{q}_x} \frac{1}{a_{\langle xx_{\bar{k}} \rangle}^2} \right) \\
&\quad -\frac{1}{d_y} \left( \frac{1}{a_{\langle xy \rangle}^2} + \sum_{k=1}^{q_y-1} \frac{1}{a_{\langle xy \rangle} a_{\langle yy_k \rangle}} + \sum_{k=1}^{\bar{q}_y} \frac{1}{a_{\langle yy_{\bar{k}} \rangle}^2} \right),
\end{aligned}
\tag{5.3}
$$

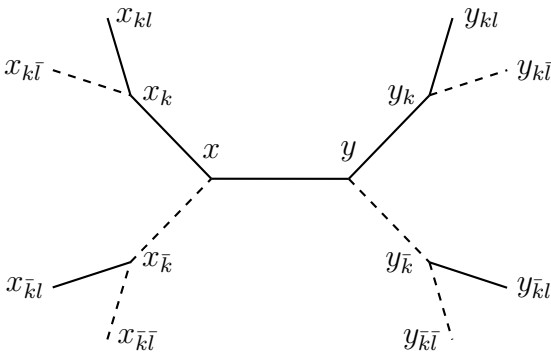

Figure 4: Barred index notation. Here edge $\langle xy \rangle$ is timelike. This configuration is for illustration purposes only, and cannot occur in the trees of Figure 3.

so that, ignoring boundary terms, the action is given by

$$S = \sum_{\langle xy \rangle, \pm} \kappa^L_{\langle xy \rangle}, \tag{5.4}$$

with the sum running over all edges.[31] Note that the third term in each bracket in Eq. (5.3) has opposite signature to $a^2_{\langle xy \rangle}$, and the action of $\mathrm{SL}_2(\mathbb{Q}_p)$ on $\kappa^L_{\langle xy \rangle}$ keeps the signature assignment of edges invariant.

The purely spacelike tree of Section 4 can be recovered by setting $q_x^+ = q_y^+ := p+1$, $q_x^- = q_y^- := 0$. This *does not* recover the curvature (4.2), but rather a curvature expression with some flipped signs. This curvature equals $-2$ for all $p$ when the edge weights are uniform, and linearizing the curvature around uniform weights recovers the linearized equations of motion $\Box j_{\langle xy \rangle} = 0$, with no vanishing prefactor at $p = 3$. Furthermore, the full nonlinear Einstein equations are precisely the ones arising from the variation of the Euclidean curvature (4.2). We thus take the point of view that for Euclidean trees the theories defined by Eqs. (4.2), (4.3), and (5.3), (5.4) are the same, at least up to boundary terms.

Let's now comment on the $q^- \neq 0$ case. $d_x$ vanishes for $q_x^+ = q_x^-$ and uniform edge lengths, and more generally for other combinations of lengths and number of neighbors also. It is not immediately obvious how to interpret what happens to the curvature in

---

[31] A subtlety present in the analysis of [45] is that the procedure used to compute the action in that paper only gives result (4.2) for large loops. For small loops, the Lipschitz extremization leading to Eq. (4.2) will in general give different expressions for the curvature. In the present paper, when small loops are present in the graph, we will remain agnostic whether Eq. (5.3) continues to apply, or whether it should be modified in some way.

this case, since the numerator also vanishes, so it is possible to take limits in certain ways. We will discuss this issue in Sections 5.4 and 5.5 below.

The rescaling $a^+_{\langle xy \rangle} \to \beta a^+_{\langle xy \rangle}$, $a^-_{\langle xy \rangle} \to \beta a^-_{\langle xy \rangle}$, with the same $\beta$ for both signatures, continues to be a symmetry of the curvature.

## 5.3 Averaging operators

When linearizing the Einstein equations we will encounter a certain type of linear operator on the graph, which we will call an *averaging* operator. Chiefly, for some functions $j_{\langle xy \rangle}$ on a graph, an averaging operator $\Box$ at edge $\langle xy \rangle$, with same signature neighbors $x^i$ and $y^j$, acts as

$$(\Box j)_{\langle xy \rangle} = \sum_{i=1}^{q_x} c_{\langle x^i x \rangle} j_{\langle x^i x \rangle} + \sum_{j=1}^{q_y} c_{\langle y^j y \rangle} j_{\langle y^j y \rangle} + c_{\langle xy \rangle} j_{\langle xy \rangle}, \tag{5.5}$$

such that

$$\sum_{i=1}^{q_x} c_{\langle x^i x \rangle} + \sum_{j=1}^{q_y} c_{\langle y^j y \rangle} + c_{\langle xy \rangle} = 0. \tag{5.6}$$

The usual Laplacian on the graph is an averaging operator. We will distinguish two types of averaging operators, named by analogy with Archimedean PDEs:

1. If $c_{\langle x^i x \rangle} > 0$, $c_{\langle y^j y \rangle} > 0$, $c_{\langle xy \rangle} < 0$ (or flipped signs), then $\Box$ is an elliptic operator.

2. If $c_{\langle x^i x \rangle} > 0$, $c_{\langle y^j y \rangle} < 0$ (or flipped signs), then $\Box$ is a hyperbolic operator.

Hyperbolic and elliptic operators on trees could in principle be defined more generally, but this definition is the bare minimum that we will need. Note, however, that it is not immediate whether these operators obey the same properties as the usual elliptic and hyperbolic operators on manifolds. Understanding this question is beyond the scope of this paper, and I will not address it here.

## 5.4 Trees of uniform edge length

Just as before, the nonlinear Einstein equations are obtained by setting the variation

$$\partial_{a_{\langle xy \rangle}} S = 0 \tag{5.7}$$

for all edges $\langle xy \rangle$. Variation (5.7) gives nonlinear equations that couple the spacelike and timelike signature edges and are quadratic in the neighbors. Uniform edge lengths $a_{\langle xy \rangle} := a$ on the entire tree are a solution to the nonlinear equations of motion for the unramified tree, and for the $p \neq 3$ totally ramified tree. For the $p = 3$ totally ramified tree, uniform edge lengths are a solution to the spacelike Einstein equations, but the timelike and horizon equations of motion on uniform edge lengths are naively indeterminate, since one of the terms they contain has vanishing numerator and denominator. However, it is possible to define an edge length $a^+$ uniform on the tree for the spacelike signature edges, and a separate edge length $a^-$ for the timelike edges, and then to take the limit $a^+ \to a^-$, which gives well-defined and vanishing equations of motion on all edges.

The configuration above, with spacelike signature edges of uniform length $a^+$, and timelike edges of uniform length $a^-$, is in fact a solution of the nonlinear equations of motion, in the following sense. For both types of tree and for all edge lengths $a^+ \neq \sqrt{w(p)}a^-$, the Einstein equations on the timelike and horizon edges are well-defined, and vanishing. Here $w(p)$ is a function of $p$ and of the type of tree. For $a^+ = \sqrt{w(p)}a^-$, the horizon and totally ramified timelike equations of motion are naively indeterminate, just as in the discussion above, however taking the limit $a^+ \to \sqrt{w(p)}a^-$ is again meaningful and gives vanishing equations of motion. The equations of motion on the spacelike edges are always well-behaved and vanishing, since they involve only $a^+$, and they are in fact the unramified tree equations of Section 4.

Function $w(p)$ is given by

$$w_r(p) = \frac{p-1}{2} \tag{5.8}$$

for the totally ramified tree, so that $p = 3$ corresponds to $w_r = 1$. For the unramified tree we have

$$w_u(p) = \frac{p^2 - p}{p + 1}. \tag{5.9}$$

The Ricci curvature for the uniform $\{a^+, a^-\}$ tree configuration is uniform,

$$\kappa^L_{\langle xy \rangle} = -2 \tag{5.10}$$

on all edges.

## 5.5 The linearized Einstein equations

We linearize the Einstein equations by writing

$$a^+_{\langle xy \rangle} = \frac{1}{\sqrt{\ell^+ + \epsilon j^+_{\langle xy \rangle}}}, \quad a^-_{\langle xy \rangle} = \frac{i}{\sqrt{\ell^- + \epsilon j^-_{\langle xy \rangle}}}, \tag{5.11}$$

for spacelike and timelike edges respectively, with $i^2 = -1$ (having $i$ in $a_{\langle xy \rangle}$ is allowed, since edges of a given signature enter $\kappa^L_{\langle xy \rangle}$ quadratically), and the coefficients $\ell^\pm$ related to the lengths $a^\pm$ in the previous section by $\ell^\pm := (a^\pm)^{-2}$. Eqs. (5.11) are linearizing the Einstein equations around a tree of uniform edge lengths $\{a^+, a^-\}$ for the timelike and spacelike signature edges, respectively. The dynamical variables are the $j^\pm_{\langle xy \rangle}$'s, and $\epsilon$ is the small expansion parameter.

The linearized equations of motion can be obtained either by expanding the curvature $\kappa^L_{xy}$ in $\epsilon$, or from the nonlinear equations coming from the action variation. To linear order in $\epsilon$ around the uniform $\{a^+, a^-\}$ tree configuration, the equations of motion for the timelike and spacelike signature edges decouple; the coupling between edges of different signature comes in only at second order.[32] Because of this, the sums in the linearized equations of motion for an edge $\langle xy \rangle$ below will always be only over edges of the same signature as $\langle xy \rangle$.

The linearized Einstein equations take the form

$$\Box j = 0 \tag{5.12}$$

for all edges and both types of trees. It is possible to write the operator $\Box$ in a "covariant" manner, that applies to all the trees and types of edges, as

$$\Box_{\langle xy \rangle} = \frac{1}{q_x \ell - \bar{q}_x \bar{\ell}} \left( -(q_x - 1)\mathbf{1}_{\langle xy \rangle} + \sum_{\substack{z \sim x \\ z \neq y}} \right) + \frac{1}{q_y \ell - \bar{q}_y \bar{\ell}} \left( -(q_y - 1)\mathbf{1}_{\langle xy \rangle} + \sum_{\substack{z \sim y \\ z \neq x}} \right), \tag{5.13}$$

where $q_{x,y}$ are the total number of edges of the same signature as $\langle xy \rangle$ at vertices $x$ and $y$ respectively, including edge $\langle xy \rangle$, and $\bar{q}_{x,y}$ are the total number of edges of opposite signature. Parameters $\ell$ and $\bar{\ell}$ take values in $\{\ell^\pm\}$, and are the inverse lengths squared of edges of the same and opposite signature as $\langle xy \rangle$. The sums run only over edges of

---

[32]I will not address in this paper whether the graviton survives at nonperturbative order. This is a question that could be asked for $\mathrm{SL}_2(\mathbb{Q}_p)$ trees also.

the same signature as $\langle xy \rangle$.

Operator (5.13) is averaging, in the sense of Section 5.3. Whether it is hyperbolic or elliptic depends on the number and types of neighbors each vertex has.

Let's now discuss the different types of edges, by using the explicit values for $q_{x,y}$ and $\bar{q}_{x,y}$ in Eq. (5.13). The spacelike linearized equations of motion are the usual unramified tree equations of motion (this is true at nonperturbative order as well), since these edges do not couple to the timelike edges. For a spacelike edge $\langle xy \rangle$ we thus have the usual graph Laplacian[33]

$$(\Box_D)_{\langle xy \rangle} = -2Q\mathbf{1}_{\langle xy \rangle} + \sum_{\substack{z \sim x \\ z \neq y}} + \sum_{\substack{z \sim y \\ z \neq x}}. \tag{5.14}$$

The linearized equations of motion for the timelike and horizon edges depend on the type of tree. For the unramified tree and a timelike edge $\langle xy \rangle$, the box operator is again just the usual graph Laplacian,

$$(\Box_{U,-})_{\langle xy \rangle} = -2p\mathbf{1}_{\langle xy \rangle} + \sum_{\substack{z \sim x \\ z \neq y}} + \sum_{\substack{z \sim y \\ z \neq x}}, \tag{5.15}$$

and for a horizon edge $\langle xy \rangle$ with endpoint $x$ neighboring the timelike edges

$$\begin{aligned}(\Box_{U,+})_{\langle xy \rangle} &= \frac{1}{(p^2 - p)\,\ell^+ - (p+1)\,\ell^-}\left(-(p^2 - p - 1)\mathbf{1}_{\langle xy \rangle} + \sum_{\substack{z \sim x \\ z \neq y}}\right) \\ &+ \frac{1}{(p^2 + 1)\ell^+}\left(-p^2\mathbf{1}_{\langle xy \rangle} + \sum_{\substack{z \sim y \\ z \neq x}}\right).\end{aligned} \tag{5.16}$$

Note the appearance of the function $w_u(p)$ introduced above. For $w_u(p)\ell^+ < \ell^-$, the sums over the neighbors of vertices $x$ and $y$ in Eq. (5.16) have opposite signs, so that the box operator in this case is hyperbolic. For $\ell^+ = \ell^-$, this corresponds to precisely the place $p = 2$. For $w_u(p)\ell^+ > \ell^-$, the box operator (5.16) is elliptic; this includes all the places $p > 2$ when $\ell^+ = \ell^-$.

When $\ell^- \to w(p)\ell^+$, the equation of motion (5.16) develops a pole. As explained above, the limit $\ell^- \to w(p)\ell^+$ makes sense formally, and the equations of motion vanish,

---

[33]In this equation and below we will not keep track of the overall normalization of the linearized operator.

provided that the coefficients of the pole and of the constant term both vanish. This gives *two* equations of motion that are first order in the neighbors,

$$-(p^2-p-1)\mathbf{1}_{\langle xy\rangle}+\sum_{\substack{z\sim x\\z\neq y}}=0,\quad -p^2\mathbf{1}_{\langle xy\rangle}+\sum_{\substack{z\sim y\\z\neq x}}=0. \tag{5.17}$$

For the totally ramified tree, the box operator for a timelike edge with vertex $x$ neighboring the spacelike edges is

$$(\Box_{R,-})_{\langle xy\rangle}=\frac{-\mathbf{1}_{\langle xy\rangle}+\mathbf{1}_{\langle x^1x\rangle^-}}{2\ell^--(p-1)\ell^+}+\frac{1}{(p+1)\ell^-}\left(-p\mathbf{1}_{\langle xy\rangle}+\sum_{\substack{z\sim y\\z\neq x}}\right), \tag{5.18}$$

with $\langle x^1x\rangle^-$ the unique timelike edge neighboring $\langle xy\rangle$. For $w_r(p)\ell^+<\ell^-$ this operator is elliptic, and for $w_r(p)\ell^+>\ell^-$ it is hyperbolic. When $\ell^+=\ell^-$ this distinction happens precisely at the place $p=3$, so that $p=2$ is elliptic and $p>3$ is hyperbolic. The $w_r(p)\ell^+=\ell^-$ case (i.e. $p=3$ if $\ell^+=\ell^-$) again makes sense formally if we take the $\ell^-\to w_r(p)\ell^+$ limit, and the two linear equations of motion we obtain are

$$\mathbf{1}_{\langle x^1x\rangle}=\mathbf{1}_{\langle xy\rangle},\quad -p\mathbf{1}_{\langle xy\rangle}+\sum_{\substack{z\sim y\\z\neq x}}=0. \tag{5.19}$$

The unique solution to the equations of motion implied by Eq. (5.19) is constant $j_{\langle xy\rangle}$.

Finally, for the horizon edges with vertex $x$ neighboring the timelike edges we have

$$\begin{aligned}(\Box_{R,+})_{\langle xy\rangle}&=\frac{1}{-2\ell^-+(p-1)\ell^+}\left(-(p-2)\mathbf{1}_{\langle xy\rangle}+\sum_{\substack{z\sim x\\z\neq y}}\right)\\&+\frac{1}{(p+1)\ell^+}\left(-p\mathbf{1}_{\langle xy\rangle}+\sum_{\substack{z\sim y\\z\neq x}}\right).\end{aligned} \tag{5.20}$$

This operator is elliptic when $w_r(p)\ell^+<\ell^-$ and hyperbolic when $w_r(p)\ell^+>\ell^-$. In the degenerate case $w_r(p)\ell^+=\ell^-$ the two linear equations of motion are

$$-(p-2)\mathbf{1}_{\langle xy\rangle}+\sum_{\substack{z\sim x\\z\neq y}}=0,\quad -p\mathbf{1}_{\langle xy\rangle}+\sum_{\substack{z\sim y\\z\neq x}}=0. \tag{5.21}$$

There is an important comment that should be made now regarding whether the

condition $w(p)\ell^+ = \ell^-$ can be fulfilled in general. The way the Einstein equations are linearized in Eq. (5.11), quantities $\ell^\pm$ correspond to edge lengths squared $(a^\pm)^2$. Thus, if we impose the restriction that Bruhat-Tits tree edge lengths should be Archimedean rationals, condition $\ell^- = w(p)\ell^+$ can only by obeyed if $w(p)$ is a square rational. It is trivial to see that this cannot happen for $w_u(p)$. In the totally ramified case primes of the form $2n^2 + 1$ satisfy the square rationality condition, the first few of which are $p = 3, 19, 73, 163, \ldots$. Whether there exist an infinite number of such primes appears to be open.

For equations linearized around $\ell^+ = \ell^-$, the types of averaging operators are summarized in Table 1. The extensions most analogous to being Lorentzian in the Archimedean sense are the totally ramified extensions, in that at almost all places the linearized Einstein equations are elliptic on the spacelike edges, and hyperbolic on the timelike edges.

|  |  | Timelike edge | Horizon edge | Spacelike edge |
|---|---|---|---|---|
| Unramified | $p = 2$ | Elliptic | Hyperbolic | Elliptic |
|  | $p > 2$ |  | Elliptic |  |
| Totally ramified | $p = 2$ | Elliptic | Hyperbolic | Elliptic |
|  | $p = 3$ | Two equations | Two equations |  |
|  | $p > 3$ | Hyperbolic | Elliptic |  |

Table 1: Types of linearized Einstein equations for different edges and both types of quadratic extensions, for $\ell^+ = \ell^-$.

## 5.6 The tree and Archimedean AdS$_2$

In this subsection I would like to draw an analogy between the quadratic extension trees and Archimedean AdS$_2$. Good reviews of AdS$_2$ can be found e.g. in [70, 71].

Lorentzian AdS$_2$ space has the topology of an infinite strip with two boundaries, and is given in global coordinates (see Figure 5) by the metric

$$ds^2 = \frac{-d\tau^2 + d\sigma^2}{\cos^2 \sigma}, \tag{5.22}$$

with $-\infty < \tau < \infty$ and $-\frac{\pi}{2} \leq \sigma \leq \frac{\pi}{2}$, so that spatial infinity corresponds to $\sigma = \pm\frac{\pi}{2}$. AdS$_2$ can be obtained from higher dimensional extremal and nonextremal black holes, however for our purposes we can think about AdS$_2$ without any a priori relation to

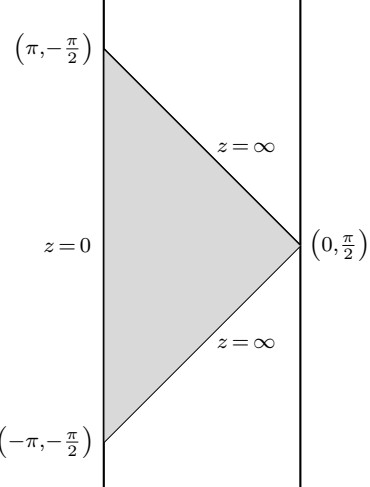

Figure 5: Global $AdS_2$ with a distinguished Poincaré wedge. The global coordinates run $-\infty < \tau < \infty$, $-\pi/2 < \sigma < \pi/2$, the Poincaré patch coordinates run $-\infty < t < \infty$, $0 < z < \infty$. We propose this spacetime as the manifold to be reconstructed from the quadratic extension trees.

higher dimensional spaces.

It is possible to define a preferred wedge on the $AdS_2$ strip (see Figure 5) by introducing a choice of coordinate time $t$ on the boundary, such that the interval $-\infty < t < \infty$ does not cover the full (left or right) boundary of the global strip, but only a region $\mathcal{A}$. The wedge $\mathcal{W}_\mathcal{C}[\mathcal{A}]$ is then simply the causal wedge of $\mathcal{A}$, i.e. the set of bulk points which can both send signals to, and receive signals from, region $\mathcal{A}$. Its horizon is the part of the boundary $\partial\mathcal{W}_\mathcal{C}[\mathcal{A}]$ which does not lie on $\mathcal{A}$.

Different choices of time correspond to different wedges. An arbitrary wedge has associated a parameter $T_H \geq 0$, the Hawking temperature of the wedge. In the $T_H \to 0$ limit, a Schwarzschild wedge becomes the Poincaré wedge, which is described by Poincaré coordinates

$$ds^2 = \frac{-dt^2 + dz^2}{z^2}. \tag{5.23}$$

The wedge has its own $\mathrm{SL}_2(\mathbb{R})$ invariance, as can be seen from the metric (5.23); a Schwarzschild wedge for $T_H > 0$ does not have this invariance, however the invariance is restored at $T_H = 0$, when the tip of the wedge on one side of the boundary touches the other side. We thus identify the solid edges in the tree as analogues of the Poincaré wedge, and the dashed edges as analogues of the complement of the wedge in global $AdS_2$. Then our dictionary is as in Table 2.

| $\mathrm{SL}_2\left(\mathbb{Q}_p\left[\sqrt{\tau}\right]\right)$ tree | Archimedean AdS$_2$ |
|---|---|
| $\mathrm{SL}_2\left(\mathbb{Q}_p\right)$ | Poincaré wedge $\mathrm{SL}_2(\mathbb{R})$ |
| $\mathrm{SL}_2\left(\mathbb{Q}_p\left[\sqrt{\tau}\right]\right)$ | Global $\mathrm{SL}_2(\mathbb{R})$ |
| Solid (timelike) edges | Interior of the Poincaré wedge |
| Dashed (spacelike) edges | Complement of the Poincaré wedge in the strip |
| Solid subtree endpoints | Poincaré wedge boundary |
| Dashed subtrees endpoints | Global minus Poincaré wedge boundary |

Table 2: Analogy between quadratic extension trees and global AdS$_2$ with a distinguished Poincaré wedge. The endpoints are points in $\mathbb{Q}_p$ and $\mathbb{Q}_p[\sqrt{\tau}] - \mathbb{Q}_p$ respectively.

# 6   Discussion

I would like to end with some speculative directions. If the $p$-adic framework is indeed relevant for Archimedean physics, as proposed in this paper, this can have consequences for open problems in the literature. Let's mention just a few.

**Cosmological constant problem:** There is a very simple way of posing the cosmological constant problem. Pick a random real number $x$ in the interval $[0, 1]$. What are the chances of getting $x$ on the order of $10^{-120}$? This is the cosmological constant problem. Of course, this way of posing the problem isn't entirely precise and admits a refinement, in that the factor of $10^{-120}$ should be obtained not at random, but presumably from a renormalization procedure that brings the Planck scale down to the cosmological constant scale. This renormalization should involve miraculous cancellations such that no additional quantum corrections of order higher than $10^{-120}$ the Planck scale are generated. It should be apparent that it is difficult to come up with such a renormalization procedure, or said another way it is difficult to obtain a scale of $10^{-120}$ without putting it in by hand.

Let's now ask if the $p$-adic point of view can improve on this situation. Although I didn't emphasize it in the rest of the paper, the adelic construction that we have implicitly used has an important feature: each adele has a cutoff prime $p_\Lambda$, above which all elements in the adele are elements of $\mathbb{Z}_p$, rather than of $\mathbb{Q}_p$. This cutoff depends on the adele, but for the purposes of this discussion let's assume the same cutoff applies to all adeles.[34] The reason the existence of this cutoff was not important for the physics discussed in this paper likely is that the models in this paper are too

---

[34]Adeles with the same cutoff still form a ring. Note that given an infinite set of adeles there need not be a cutoff that applies to all of them, so this condition is nontrivial.

simple to pick up on it. However, physical mechanisms could *in principle* depend on $p_\Lambda$. Which brings us to the punchline: $p_\Lambda$ can act as a scale that adelic realizations of physics are automatically equipped with. What is the natural order of magnitude of this scale? In the Archimedean case, a random real in the $[0, \ldots, 1]$ interval multiplying $\Lambda_{\text{Planck}}$ will naturally be $O(1)$. In contrast, the set of primes is not bounded from above, so the natural value of $p_\Lambda$ is arbitrarily large. From this point of view, a factor such as $10^{120}$ may even appear too small.[35]

Of course, this proposal does not *solve* the cosmological constant problem. To do so, we would also need to construct a physical mechanism which takes advantage of the scale $p_\Lambda$ to set the cosmological constant scale. It will be interesting to investigate if this can be achieved in string theory.

**Black hole evaporation:** If there exists a theory of quantum gravity on the $p$-adic side, it should be expected that it will be unitary, either at every $p$ or when all places are considered together. It should be possible to consider black hole evaporation in this unitary theory, and to recover all information from the Hawking radiation, assuming no remnants. However, this unitary evolution on the $p$-adic side need not map to an unitary evolution on the Archimedean side. Some "projection" operation could occur when passing from the finite places to the Archimedean one, so that the effective Archimedean theory is not unitary (this could be engineered most strongly e.g. if some of the physics needed in describing the quantum aspects of Hawking evaporation has no Archimedean interpretation). If this happens, then black hole evaporation at the Archimedean place will look like information loss, even if the underlying $p$-adic physics is unitary.

Various types of black hole solutions exist in various theories, and not all may exhibit this severe behavior where Archimedean evolution is not unitary. Even in cases of unitary evolution on the Archimedean side, $p$-adic mechanisms could ensure novel ways of encoding correlations, and help soften the black hole information paradox.

**Tensor networks:** It has recently been pointed out that, in the usual (Archimedean) context of AdS/CFT, bulk reconstruction from the boundary CFT data needs to exhibit error correcting properties, otherwise the AdS/CFT proposal is inconsistent with field theory [73]. There are currently different proposals in the literature for how this quantum error correction could be implemented in practice, such as operator reconstruction approaches related to modular flow [74–77] and tensor network techniques [78–80].

---

[35]This point of view on the cosmological constant was first pointed out in [72].

While operator reconstruction seems to be consistent with field theory and holography in the regimes where it has been analyzed, tensor network approaches have difficulties with what essentially are the symmetries of the bulk (although the manner these difficulties are usually stated in the literature is slightly different). Relatedly, it is not clear how to make the connection between tensor networks and Lorentzian time evolution in the bulk, and how to describe time-evolving tensor networks as models of bulk gravity.[36]

The Bruhat-Tits building framework discussed in this paper provides a natural way of connecting gravity to tensor networks, in the context of holography. The key insight is that difficulties related to symmetry can be avoided if the tensor networks naturally live at the finite places, rather than at the Archimedean place. This was already pointed out by [44], however we are now in a position to make a precise proposal for how this construction could work. Let's consider Lorentzian $AdS_3/CFT_2$ on the Archimedean side. From the general discussion in Section 3, we expect the $p$-adic side to be given by $\mathrm{BT}\,(\mathrm{SL}_2 \times \mathrm{SL}_2)$, with the $\mathrm{SL}_2$'s acting on $\mathbb{Q}_p$ or its quadratic extensions, such that the building has a Lorentzian interpretation. There should then exist a natural notion of Euclidean sections in this building, with the tensor network vertices living on the facets of these sections. In this setting the tensor networks only need to be compatible with the discrete symmetries of the Euclidean sections, rather than with the continuous symmetries of an Archimedean space.[37]

Another feature of this approach is that since the building comes from number-theoretic data, it may be possible to use the same data to construct the tensor network codes. Some steps in this direction have already been taken in [81].

**Black hole microstates:** The examples we have discussed represent simple models in which Archimedean physics emerges from $p$-adic physics. Ultimately, a fundamental question is whether $p$-adic decomposition will indeed turn out to have anything to do with the degrees of freedom of quantum gravity, as it has been proposed here. It will

---

[36]One could take the point of view that tensor networks are supposed to only be a toy model of holography, which will never recover *all* the properties of bulk asymptotically AdS spaces. However, in this paragraph I would like to nonetheless push for a stronger point of view, and ask whether precise contact with gravitational dynamics can be made.

[37]Progress in connecting bulk tensor network and HKLL reconstructions in $p$-adic settings has been made by [47], which considers tensor networks that coincide with the Bruhat-Tits trees, i.e. analogues of Archimedean $AdS_2/CFT_1$. In [44] and the proposal in this section the vertices of the tensor network live on facets and the tensor network connection cut across the edges of the building (or of the tree), so in this sense these proposals are analogues of $AdS_3/CFT_2$.

be interesting to investigate whether the more advanced objects mentioned and hinted at in this paper could help make progress in this direction.

**Acknowledgments.** I would like to thank Steven Gubser, Matthew Headrick, Jennifer Lin, Maria Nastasescu, and Sarthak Parikh for valuable discussions and feedback on early versions on this draft. I would furthermore like to thank Ahmed Almheiri, Pawel Caputa, Harsha Hampapura, Daniel Harlow, Tom Hartman, Matthew Heydeman, Christian Jepsen, Isaac Kim, Nima Lashkari, Hong Liu, Matilde Marcolli, Cheng Peng, Djordje Radicevic, Ingmar Saberi, Washington Taylor, Brian Trundy, and Barton Zwiebach for useful discussions. This work was supported in part by the Simons Foundation, by the U.S. Department of Energy under grant DE-SC-0009987, and by a grant from the Brandeis University Provost Office. I would like to thank the Stanford Institute for Theoretical Physics at Stanford University, the Center for Theoretical Physics at the Massachusetts Institute of Technology, the Center of Mathematical Sciences and Applications at Harvard University, and the Okinawa Institute for Science and Technology for hospitality. This work was performed in part at the Aspen Center for Physics, which is supported by National Science Foundation grant PHY-1607761.

# A $p$-adic numbers, basic integration

In these Appendices I will present some results on $p$-adic numbers. This review is not comprehensive, and more details can be found e.g. in [43, 50, 55, 82, 83].

A $p$-adic number $x \in \mathbb{Q}_p$ can be represented as a digit expansion

$$x = p^{v(x)} \sum_{i=0}^{\infty} x_i p^i, \tag{A.1}$$

where $v(x) \in \mathbb{Z}$, $x_0 \neq 0$ and $x_i \in \{0, \ldots, p-1\}$. Addition and multiplication of $p$-adic numbers is performed in the usual way (with carry). The norm of $x$ is defined as

$$|x|_p := p^{-v(x)}, \tag{A.2}$$

and $v(x)$ is called the valuation. It can be checked that Definition (A.2) obeys the norm axioms.

It is customary to introduce $p$-adic units, integers and maximal ideal as

$$\mathbb{U}_p \; := \; \left\{ x \in \mathbb{Q}_p : |x|_p = 1 \right\}, \tag{A.3}$$

$$\mathbb{Z}_p \; := \; \left\{ x \in \mathbb{Q}_p : |x|_p \leq 1 \right\}, \tag{A.4}$$

$$\mathfrak{p} \; := \; \left\{ x \in \mathbb{Q}_p : |x|_p < 1 \right\}. \tag{A.5}$$

In this paper integration on $\mathbb{Q}_p$ is always done with the translation-invariant Haar measure, so that

$$d(x + a) = dx. \tag{A.6}$$

Under a change of variables $x \to f(x)x$ the integration measure changes by $|f(x)|_p$, and volumes are normalized so that

$$\int_{\mathbb{Z}_p} 1 = 1. \tag{A.7}$$

There exists also a multiplication-invariant measure on $\mathbb{Q}_p$, but it is not used in this paper.

# B   Characters, Fourier transforms, Gamma functions

This Appendix introduces additive and multiplicative characters on $\mathbb{Q}_p$, and related concepts. Since the multiplicative characters $\pi_{s,\tau}$ that will be defined here depend on the sign characters of $\mathbb{Q}_p$, this Appendix would have followed most naturally after Appendix C below. However, to streamline presentation it has been placed before Appendix C, with the understanding that the sign functions $\mathrm{sgn}_\tau(x)$ that will be needed are themselves multiplicative characters on $\mathbb{Q}_p$, that depend on a parameter $\tau \in \mathbb{Q}_p$ and take values in $\{\pm 1\}$. The precise definition of these characters is not necessary for Appendix B, and will be given in Appendix C.

## B.1   Characters on $\mathbb{Q}_p$

A $p$-adic additive character $\chi(x)$ is a continuous function

$$\chi : \mathbb{Q}_p \to \mathbb{C}^\times \tag{B.1}$$

such that

$$\chi(x+y) = \chi(x)\chi(y) \tag{B.2}$$

for all $x, y \in \mathbb{Q}_p$. The canonical choice of an additive character is

$$\chi(x) = e^{2\pi i \{x\}_p}, \tag{B.3}$$

with $\{x\}_p$ the fractional part of $x$. Here the fractional part is an instruction to remove all positive $p$ powers in the expansion of $x$, i.e.[38]

$$\left\{ \sum_{i=m}^{\infty} a_i p^i \right\}_p = \sum_{i=m}^{-1} a_i p^i, \tag{B.4}$$

for $m < 0$ and $a_i \in \{0, \ldots, p-1\}$, with the right-hand side taken to be an element of $\mathbb{Q}$. If $m \geq 0$, the fractional part is defined to be zero.

Multiplicative characters are continuous functions

$$\pi : \mathbb{Q}_p^\times \to \mathbb{C} \tag{B.5}$$

such that

$$\pi(xy) = \pi(x)\pi(y) \tag{B.6}$$

for all $x, y \in \mathbb{Q}_p^\times$. Canonical multiplicative character choices are[39]

$$\pi_{s,\tau}(x) = |x|^s \operatorname{sgn}_\tau x, \tag{B.7}$$

$$\pi_s(x) = |x|^s. \tag{B.8}$$

with the sign function $\operatorname{sgn}_\tau x$ defined in Appendix C.

---

[38]Strictly speaking, the fractional part notation in Eq. (B.3) can be omitted, with the understanding that integers in the exponential in Eq. (B.3) drop out, as is usually the case with complex exponentials. However, in this paper I keep the $\{\cdot\}_p$ notation, as a reminder that the resulting series also passes from $\mathbb{Q}_p$ to $\mathbb{C}$.

[39]It is in fact not hard to give a complete characterization of all additive and multiplicative characters of $\mathbb{Q}_p$. See [55] for the details.

## B.2 Fourier transforms and other integral identities

The $p$-adic Fourier transform is defined as

$$f(k) := \int \chi(kx) f(x), \tag{B.9}$$

and the normalization for the inverse Fourier transform can be chosen to be

$$f(x) = \int \chi(-kx) f(k). \tag{B.10}$$

These formulas function just as in the Archimedean case, with the understanding that $\chi(x)$ corresponds to the usual factor of $e^{2\pi i x}$.

The Dirac delta function has a representation

$$\delta(x) = \int \chi(kx). \tag{B.11}$$

The $p$-adic Gaussian integral is reminiscent of the Archimedean one, and is given by [3, 10] (see also [7])

$$\int \chi\left(ax^2 + bx\right) = \frac{\lambda_{(p)}(a)}{|2a|^{\frac{1}{2}}} \chi\left(-\frac{b^2}{4a}\right), \tag{B.12}$$

where $|2| = 1$ for $p > 2$ and $|2| = 1/2$ at $p = 2$. Here $\lambda_p(a)$ is, up to a phase factor, a Legendre symbol, i.e. for $p > 2$ we have

$$\lambda_{(p)}(a) = \begin{cases} 1 & \text{if } v(a) \text{ is even} \\ (a_0|p) & \text{if } v(a) \text{ is odd, } p = 1 \bmod 4 \\ i(a_0|p) & \text{if } v(a) \text{ is odd, } p = 3 \bmod 4 \end{cases}, \tag{B.13}$$

with $v(a)$ and $a_0$ as in the $p$-adic digit decomposition (A.1), and $(a_0|p)$ the Legendre symbol defined in Appendix C.2 below. At $p = 2$ the prefactor is instead

$$\lambda_{(2)}(a) = \begin{cases} \frac{1}{\sqrt{2}} \left[ 1 + i(-1)^{a_1} \right] & \text{if } v(a) \text{ is even} \\ \frac{1+i}{\sqrt{2}} i^{a_1} (-1)^{a_2} & \text{if } v(a) \text{ is odd} \end{cases}, \tag{B.14}$$

with $a_{1,2}$ 2-adic digits in (A.1), and we remember that $a_0 = 1$.

Although $p$-adic integration is simple, deriving relations such as Eq. (B.12) can be

somewhat involved, if straightforward.

## B.3 Gamma functions

The Gel'fand–Graev Gamma function can be defined as a function of multiplicative characters, as [55]

$$\Gamma(\pi) := \int \chi(x)\pi(x)|x|^{-1}. \tag{B.15}$$

It has the following properties:

1. The only singular point is at $\pi_0(x) = 1$.

2. The only zero is at $\pi_1(x) = |x|$.

3. It obeys a functional equation,

$$\Gamma(\pi)\Gamma(\pi_1\pi^{-1}) = \pi(-1). \tag{B.16}$$

A useful identity is the Fourier transform of a multiplicative character $\pi(t)$,

$$\int \chi(kx)\pi(x) = \frac{\Gamma(\pi\pi_1)}{\pi(k)\pi_1(k)}, \tag{B.17}$$

with $\pi_1$ defined by Eq. (B.8). Note that setting $\pi(x) = 1$ in Eq. (B.17) corresponds to the zero of the Gamma function on the RHS, which is compatible with Eq. (B.11), since Eq. (B.17) assumes $|k| \neq 0$.

Explicit expressions for the Gamma functions for all $\tau$ equivalence classes and fields $\mathbb{Q}_p$ for $p > 2$, and $\mathbb{R}$ can be found in Table 3. Here $\theta_\tau$ are all four 4th order roots of unity in $\mathbb{C}$, depending on the values of $p \mod 4$ and $\tau$, with the property that

$$\theta_\tau^2 = \text{sgn}_\tau(-1), \tag{B.18}$$

and $\Gamma(s)$ appearing in the Archimedean place expression is the usual Euler Gamma function.

| $\tau$ | $\mathbb{Q}_p$ | | | $\mathbb{R}$ | |
|---|---|---|---|---|---|
| | $1$ | $\epsilon$ | $p,\ \epsilon p$ | $-1$ | $1$ |
| $\Gamma(\pi_{s,\tau})$ | $\frac{1-p^{s-1}}{1-p^{-s}}$ | $\frac{1+p^{s-1}}{1+p^{-s}}$ | $\theta_\tau p^{s-\frac{1}{2}}$ | $i2^{1-s}\pi^{-s}\sin\left(\frac{\pi s}{2}\right)\Gamma(s)$ | $2^{1-s}\pi^{-s}\cos\left(\frac{\pi s}{2}\right)\Gamma(s)$ |

Table 3: Gamma functions for various fields and values of $\tau$. For more details see [55], but beware typos.

# C  Quadratic extensions and sign characters

## C.1  Quadratic extensions

This section is a quick review of field extensions of $L \coloneqq \mathbb{Q}_p$, focusing on quadratic extensions. A quadratic extension is a field $K \coloneqq \mathbb{Q}_p[\sqrt{\tau}]$ with $\tau$ a square-free element in $\mathbb{Q}$, notation $K/L$. An element in $K$ is of the form

$$x \in K \quad \Leftrightarrow \quad x = a + b\sqrt{\tau}, \quad a, b \in \mathbb{Q}_p. \tag{C.1}$$

In general, an extension $K$ of degree $n = [K/L]$ of a field $L$ is a field that is an $n$-dimensional vector space over $L$ that also contains $L$, but for the purposes of a definition it suffices to demand that $L$ is a subfield of $K$, meaning that it closed under the field operations. Of course, for quadratic extensions $n = 2$.

**Remark 1** *Two quadratic extensions $L[\sqrt{x}]$, $L[\sqrt{y}]$ of a disconnected field $L$ are equivalent iff $xy^{-1}$ is a square in $L$ (p. 131 of [55]).*

We would like to define a norm $|\cdot|_K : K \to \mathbb{R}_{\geq 0}$ on the extension $K/\mathbb{Q}_p$, obeying the properties

1. $|x|_K = 0$ iff $x = 0$.

2. $|xy|_K = |x|_K |y|_K$ for $x, y \in K$.

3. $|x + y|_K \leq |x|_K + |y|_K$ for $x, y \in K$.

4. $|x|_K = |x|_p$ for $x \in \mathbb{Q}_p$.

**Theorem 1** *For $K/L$ a finite extension, there exists a unique extension $|\cdot|_K$ of $|\cdot|_L$ to $K$.*

In order to define the norm on $K/\mathbb{Q}_p$, we need to introduce the norm map $N_{K/L}(x)$. Some good references on the norm map are Chapter 5 of [82], but a succinct discussion can be found in [43], which we will mostly follow. Fix any element $x \in K$, and introduce the map

$$y \to xy \tag{C.2}$$

for any $y \in K$. This induces a linear map on $L^n$, so we can take its determinant $N_{K/L}(x)$. This determinant is the norm map from $K$ to $L$,

$$N_{K/L} : K \to L. \tag{C.3}$$

The norm map exhibits some nice properties [82]:

1. For an element $x \in L$ and an extension $K/L$ of degree $n$, we have

$$N_{K/L}(x) = x^n. \tag{C.4}$$

2. It is multiplicative. For $x, y \in K$,

$$N_{K/L}(xy) = N_{K/L}(x)N_{K/L}(y). \tag{C.5}$$

Using the norm map, we can define the norm on $K$ as

$$|x|_K := \sqrt[n]{|N_{K/L}(x)|_L}. \tag{C.6}$$

Note that this definition makes sense, since the norm map takes values in $L$.

**Remark 2** *For a quadratic extension $K/\mathbb{Q}_p$, for an element $x \in K$ of the form*

$$x = a + b\sqrt{\tau} \tag{C.7}$$

*we can define its conjugate as*

$$\bar{x} := a - b\sqrt{\tau} \tag{C.8}$$

*and the norm map is*

$$N(x) = x\bar{x} = a^2 - b^2\tau, \tag{C.9}$$

*so that the norm on $K$ is*

$$|x|_K = \sqrt{|a^2 - b^2\tau|_p}. \tag{C.10}$$

I would now like to describe the extensions $K/\mathbb{Q}_p$ a bit better. To do this, let's introduce the ramification index $e$. First, remember that for $x \in K$, the $p$-adic valuation $v_p(x)$ is defined as

$$|x|_K := p^{-v_p(x)}, \tag{C.11}$$

and $v_p(0) = \infty$. One way to define the ramification index $e$ is as in Proposition 5.4.2 in [82], which I promote to a theorem:

**Theorem 2** *The image of $v_p$ in $\mathbb{Q}$ is of the form $\frac{1}{e}\mathbb{Z}$, with $e$ dividing $n = [K/\mathbb{Q}_p]$.*

Another way to state the definition of the ramification index is that $e$ is the smallest divisor of $n$ such that $|x|_K^e$ is an integer power of $p$ for all $x \in K$ [43]. It is standard notation to also introduce

$$f = f(K/\mathbb{Q}_p) = \frac{n}{e}. \tag{C.12}$$

An element $\mathfrak{p} \in K$ is called a uniformizer if

$$v_p(\mathfrak{p}) = \frac{1}{e}. \tag{C.13}$$

For an extension $K/\mathbb{Q}_p$ there generically exist many uniformizers.

Just as for $\mathbb{Q}_p$, units, a valuation ring, and its maximal ideal[40] can be introduced,

$$\mathbb{U}_K := \{x \in K : |x|_K = 1\}, \tag{C.14}$$

$$\mathbb{Z}_K := \{x \in K : |x|_K \leq 1\}, \tag{C.15}$$

$$\mathfrak{p}_K := \{x \in K : |x|_K < 1\}. \tag{C.16}$$

Note that the maximal ideal is generated by the uniformizer $\mathfrak{p}$. Since $\mathfrak{p}_K$ is a maximal ideal in the valuation ring, the quotient $\mathbb{Z}_K/\mathfrak{p}_K$ is a field, called the residue field **k**:

$$\mathbf{k} = \mathbb{Z}_K/\mathfrak{p}_K. \tag{C.17}$$

**Theorem 3** *The residue field is the finite field with $p^f$ elements, where $f = n/e$ as in Eq. (C.12) above:*

$$\mathbf{k} = \mathbb{F}_{p^f}. \tag{C.18}$$

---

[40]Beware of notational clash with the uniformizer $\mathfrak{p}$.

A proof of this theorem can be found e.g. in [82].

Any $x \in K$ has a digit decomposition in terms of the uniformizer and elements $a_i$ of the residue field as [43],

$$x = \mathfrak{p}^{v_K(x)} \sum_{i=0}^{\infty} a_i \mathfrak{p}^i, \quad a_0 \neq 0. \tag{C.19}$$

It is possible to make this decomposition more detailed, by writing it as [50]

$$x = \mathfrak{p}^{v_K(x)} \epsilon^{w(x)} a(x), \tag{C.20}$$

where $\epsilon$ is a generator of $\mathbb{F}_{p^f}^{\times}$, $w(x) \in \{1, \ldots, p^f\}$, and $a(x)$ is some element in the multiplicative group

$$A = \{a \in K : |a - 1| < 1\}. \tag{C.21}$$

Once $\mathfrak{p}$ and $\epsilon$ are fixed, decomposition (C.20) is unique. This decomposition will be crucial for writing sign characters explicitly in Section C.2 below.

There are three types of extensions of $\mathbb{Q}_p$:

1. If $e = 1$, then the extension $K/\mathbb{Q}_p$ is the unique unramified extension. In this case it is possible (and customary) to choose the uniformizer $\pi = p$. This extension can be obtained by adjoining $\tau = \epsilon$ a primitive $(p^n - 1)$-th root of unity to $\mathbb{Q}_p$ (see e.g. Proposition 5.4.11 in [82] for more details).

2. If $e > 1$, then $K/\mathbb{Q}_p$ is ramified.

3. If $e = n$, then $K/\mathbb{Q}_p$ is totally ramified, and we can choose the uniformizer $\pi = p^{1/n}$.

When considering quadratic extensions $n = 2$, so the only possibilities are unramified or totally ramified extensions.

The quadratic extensions of $\mathbb{Q}_p$ can be classified as follows:

**Theorem 4** *For $p > 2$, there are three distinct quadratic extensions of $\mathbb{Q}_p$, which can be taken to be $\mathbb{Q}_p[\sqrt{\epsilon}]$ (unramified), $\mathbb{Q}_p[\sqrt{p}]$ (totally ramified), and $\mathbb{Q}_p[\sqrt{\epsilon p}]$ (totally ramified), where $\epsilon$ is a primitive $(p^2 - 1)$-th root of unity, just as before.*

For a proof see e.g. [55]. This is the same as saying that the cosets of $\mathbb{Q}_p^{\times}/\left(\mathbb{Q}_p^{\times}\right)^2$ have representatives $\{1, \epsilon, p, \epsilon p\}$.

**Theorem 5** *For $p = 2$, there are seven distinct quadratic extensions of $\mathbb{Q}_2$, which can be taken to be $\mathbb{Q}_2[\sqrt{\tau}]$ with $\tau = -1, \pm 2, \pm 3, \pm 6, \pm 10$. An alternate choice is $\tau = 2, 3, 5, 6, 7, 10, 14$.*

## C.2 Quadratic (sign) characters

In this section we will introduce sign functions for arbitrary fields $K$.

A quadratic residue mod $n$ is an integer $q$ such that

$$x^2 = q \mod n \tag{C.22}$$

for some integer $x$. An integer $q$ which is not a quadratic residue is a quadratic nonresidue. The Legendre symbol is defined as

$$(a|p) := \begin{cases} +1 & a \text{ is a quadratic residue mod } p \text{ and } a \neq 0 \mod p \\ -1 & a \text{ is a quadratic nonresidue mod } p \\ 0 & a = 0 \mod p \end{cases} \tag{C.23}$$

Via Euler's criterion, this is the same as

$$(a|p) = a^{\frac{p-1}{2}} \mod p. \tag{C.24}$$

For a field $K$, a multiplicative character is a function $\pi : K^\times \to \mathbb{C}$ such that $\pi(xy) = \pi(x)\pi(y)$. Quadratic characters are multiplicative characters $\pi$ such that $\pi^2 = 1$, i.e. such that $\pi(x) = \pm 1$ for all $x \in K^\times$.

There are three (equivalent) ways of defining the sign functions:

1. With the help of squares in $K$, as in Eq. (C.25) below.

2. From the decomposition (C.20).

3. As Hilbert symbols.

This Appendix will discuss all three ways, first presenting some general properties (for the first and third definitions) and then moving to a field-by-field basis.

Let's start with the first definition. For any $\tau \in K^\times$, the sign function is a quadratic

character $\text{sgn}_\tau x$ defined as [55]

$$\text{sgn}_\tau x := \begin{cases} +1, & x = a^2 - \tau b^2 \text{ for some } a, b \in K \\ -1, & x \neq a^2 - \tau b^2 \ \forall \ a, b, \in K \end{cases} . \tag{C.25}$$

From this formula $\text{sgn}_\tau = \text{sgn}'_\tau$ if $\tau/\tau'$ is a square in $K$, similarly to Remark 1. Eq. (C.25) implies that the sign functions for $K$ are classified by the coset representatives of the quotient $K^\times/(K^\times)^2$, so there is a bijection between the nontrivial sign characters and the quadratic extensions of $K$.

Let's now switch to the Hilbert symbol definition. It turns out there is a strong connection between the sign functions defined on a field $L$ and quaternion algebras defined on $L$, as explained by [83] (compare also with the formulas of Section 2 of [5,20]). For a field $L$, the quaternion algebra $D_{L,a,b}$ with $a, b \in L$ is defined as

$$D_{L,a,b} := L\langle \mathbf{i}, \mathbf{j} \rangle / \left( \mathbf{i}^2 + 1, \mathbf{j}^2 + 1, \mathbf{ij} + \mathbf{ji} \right) . \tag{C.26}$$

A natural question to ask is when $D_{L,a,b}$ is isomorphic to $M_2(L)$, the algebra of $2 \times 2$ matrices with entries in $L$. This is answered by [83], in the form of the following theorem:

**Theorem 6** *If $a \in L^\times$ is a square in $L^\times$, then for any $b \in L^\times$, $D_{L,a,b}$ is isomorphic to $M_2(L)$. If $a \in L^\times$ is not a square, then $D_{L,a,b}$ is isomorphic to $M_2(L)$ iff $b \in L^\times$ can be written as $b = x^2 - ay^2$, for some $x, y \in L$.*

Theorem 6 classifies quaternion algebras. Of course, this classification is the same as that of the quadratic extensions above, and of the sign functions below.

Suppose now $L$ is either $\mathbb{R}$ or $\mathbb{Q}_p$. Then the Hilbert symbol $(a, b)$ for $a, b \in L$ is defined as

$$(a, b) := \begin{cases} +1, & D_{L,a,b} \text{ is isomorphic to } M_2(L) \\ -1, & D_{L,a,b} \text{ is not isomorphic to } M_2(L) \end{cases} . \tag{C.27}$$

The punchline is that for quadratic extensions the Hilbert symbol is the same as the sign function,

$$\text{sgn}_a b = (a, b). \tag{C.28}$$

With $a, b, c, r, s \in L^\times$, the Hilbert symbol obeys nice properties [83]:

- $(a, b) = (b, a) = (a, -ab), \quad (a, b) = (ar^2, bs^2).$

- Detection of squares: $(a, b) = 1$ for all $a$ iff $b$ is a square in $L$.

- Bilinearity: $(a, bc) = (a, b)(a, c)$.

- Symbol properties: $(a, 1 - a) = 1$ for $a \neq 1$, $(a, -a) = 1$ for all $a$.

The following very nice result is due to Hilbert:

**Theorem 7** *For fixed rationals $a$, $b$ in $L$ (assume that $L = \mathbb{Q}_p$, else the places refer to the primes of the number field):*

- $(a, b)_p = 1$ *at almost all places $p$.*

- $\prod_v (a, b)_v = 1$.

## C.3 Quadratic characters on a field-by-field basis

### C.3.1 Quadratic characters for $\mathbb{R}^\times$

From Eq. (C.25) there are precisely two quadratic characters for $\mathbb{R}^\times$ [6,20]: the trivial character,

$$\text{sgn}_1 x = +1 \quad \forall\ x \in \mathbb{R}^\times, \tag{C.29}$$

and the usual sign function

$$\text{sgn}_{-1} x = \begin{cases} +1 & x > 0 \\ -1 & x < 0 \end{cases} \quad \forall\ x \in \mathbb{R}^\times. \tag{C.30}$$

### C.3.2 Quadratic characters for $\mathbb{C}^\times$

The only sign character is the trivial one, since any polynomial equation in $\mathbb{C}$ has solutions.

### C.3.3 Quadratic characters for $\mathbb{F}_p^\times$

Pick $p > 2$ so the field is nontrivial. There are exactly two characters [50]: the trivial character, and the Legendre symbol $(x|p)$, which can be even or odd since

$$\text{sgn}(-1) = (1|p) = (-1)^{\frac{p-1}{2}}. \tag{C.31}$$

Note that $\mathrm{sgn}(-1) = 1$ if $p = 1 \mod 4$ and $\mathrm{sgn}(-1) = -1$ if $p = 3 \mod 4$. This is a recurring theme.

### C.3.4  Quadratic characters for $\mathbb{Q}_p^\times$, $p > 2$

For field extensions $L$ with residue field of odd characteristic, the coset representatives of $L^\times / (L^\times)^2$ are given by

$$\{1, \epsilon, \mathfrak{p}, \epsilon\mathfrak{p}\}, \tag{C.32}$$

so that for $L = \mathbb{Q}_p$ we can make the canonical uniformizer choice $\mathfrak{p} = p$.[41]

Using decomposition (C.20), the sign function $\rho(x)$ for $x \in L$ can be written explicitly. Since $\mathrm{sgn}\, a(x) = 1$ [50], only the signs of the uniformizer $\mathfrak{p}$ and root of unity $\epsilon$ matter for the sign function $\rho(x)$; denoting $\sigma_\mathfrak{p} := \mathrm{sgn}\,\mathfrak{p}$, $\sigma_\epsilon := \mathrm{sgn}\,\epsilon$, from decomposition (C.20) we have

$$\rho_{\sigma_\mathfrak{p}, \sigma_\epsilon}(x) = \sigma_\mathfrak{p}^{v(x)} \sigma_\epsilon^{w(x)}, \tag{C.33}$$

with the choices for $\sigma_\mathfrak{p}$ and $\sigma_\epsilon$ determined by the coset representatives in (C.32), as in Table 4 (see [50] for the details). It is important to remark that we must have $p = 3 \mod 4$ in order for $\mathrm{sgn}_\tau(-1)$ to be nontrivial, and this only happens for the totally ramified extensions. Note that Eq. (C.33) applies when $L$ is either $\mathbb{Q}_p$ or an extension of $\mathbb{Q}_p$, $p > 2$.

We now restrict to $L = \mathbb{Q}_p$, $p > 2$. Decomposing $x$ and $\tau$ as

$$x = p^{v(x)} \sum_{i=0}^\infty x_i p^i, \ x_0 \neq 0, \tag{C.34}$$

$$\tau = p^{v(x)} \sum_{i=0}^\infty \tau_i p^i, \ \tau_0 \neq 0, \tag{C.35}$$

there is an explicit formula for the sign function in terms of Legendre symbols [5, 20],

$$\mathrm{sgn}_\tau x = (x_0|p)^{v(\tau)} (\tau_0|p)^{v(x)} (-1|p)^{v(\tau)v(x)}. \tag{C.36}$$

This is also the formula one obtains by thinking of the sign function as a Hilbert symbol $(\tau, x)$ (compare with Eq. (2.2.1) in [83]).

---

[41]In general, the sign function does not depend on the choice of the root $\epsilon$ in the decomposition (C.20), but changing the uniformizer can cause a shift. However, I will keep $\mathfrak{p} = p$ fixed.

| $\tau$ | 1 | $\epsilon$ | $p$ | $\epsilon p$ |
|---|---|---|---|---|
| $\sigma_p$ | 1 | $-1$ | $(-1\|p)$ | $-(-1\|p)$ |
| $\sigma_\epsilon$ | 1 | 1 | $-1$ | $-1$ |
| $\mathrm{sgn}_\tau(-1)$ | 1 | 1 | $(-1\|p)$ | $(-1\|p)$ |

Table 4: The sign functions on $\mathbb{Q}_p$, $p > 2$, with Legendre symbol $(-1|p) = (-1)^{\frac{p-1}{2}}$. Table taken from [50].

### C.3.5  Quadratic characters for $\mathbb{Q}_2$

Following [50], for $x \in \mathbb{Q}_2$ expanded as[42]

$$x = 2^{v(x)} (1 + 2x_1 + 4x_2 + \dots) \tag{C.37}$$
$$:= 2^{v(x)} a_x,$$

the sign function is

$$\rho_{\sigma_{\mathfrak{p}}, \sigma_{x_1}, \sigma_{x_2}}(x) = \sigma_{\mathfrak{p}}^{v(x)} \sigma_{x_1}^{x_1} \sigma_{x_2}^{x_2}. \tag{C.38}$$

The sign choices of the various $\sigma$'s are determined by the coset representatives, as in Table 5.

In terms of the Legendre symbols, introducing the decomposition

$$\tau = 2^{v(\tau)} (1 + 2\tau_1 + 4\tau_2 + \dots), \tag{C.39}$$
$$:= 2^{v(\tau)} a_\tau,$$

we have [5, 20]

$$\mathrm{sgn}_\tau x = (-1)^{\tau_1 x_1 + (\tau_1 + \tau_2)v(x) + (x_1 + x_2)v(\tau)}, \tag{C.40}$$

and from Hilbert symbol considerations we obtain [83]

$$\mathrm{sgn}_\tau x = (-1)^{\frac{(a_x - 1)(a_\tau - 1)}{4}} (-1)^{v(x)\frac{a_\tau^2 - 1}{8} + v(\tau)\frac{a_x^2 - 1}{8}}, \tag{C.41}$$

which is the same as Eq. (C.40).

---

[42]We are setting the uniformizer $\mathfrak{p} = 2$, as usual.

| $\tau$ | 1 | $-1$ | 2 | $-2$ | 3 | $-3$ | 6 | $-6$ |
|---|---|---|---|---|---|---|---|---|
| $\sigma_2$ | 1 | 1 | 1 | 1 | $-1$ | $-1$ | $-1$ | $-1$ |
| $\sigma_{a_1}$ | 1 | $-1$ | $-1$ | 1 | $-1$ | 1 | 1 | $-1$ |
| $\sigma_{a_2}$ | 1 | 1 | $-1$ | $-1$ | 1 | 1 | $-1$ | $-1$ |
| $\mathrm{sgn}_\tau(-1)$ | 1 | $-1$ | 1 | $-1$ | $-1$ | 1 | $-1$ | 1 |

Table 5: The sign functions on $\mathbb{Q}_2$. Table taken from [50].

# D    Adeles and product formulas

## D.1    Adeles

The treatment in this appendix will be schematic, and for more details the reader is referred to [55]. An adele is an infinite sequence

$$a := (a_\infty, a_2, \ldots, a_p, \ldots) \tag{D.1}$$

with $a_\infty \in \mathbb{R}$, $a_p \in \mathbb{Q}_p$, and with the condition that all $a_{p \geq P}$ are $p$-adic integers, with $P$ an arbitrary (possibly large) prime that depends on $a$.

One The set of adeles is a ring under componentwise addition and multiplication, called the ring of adeles $\mathbb{A}$. An additive character on the adeles is defined as

$$\chi_a(r) := e^{-2\pi i a_\infty r} \prod_p e^{2\pi i \{a_p r\}}, \tag{D.2}$$

with $a \in \mathbb{A}$ and $r \in \mathbb{Q}$. The factors in the product on the right-hand side of Eq. (D.2) are precisely the additive characters on $\mathbb{R}$ and $\mathbb{Q}_p$, and the meaning of the fractional part is as in Eq. (B.4).

One implication of the following theorem is useful for quantum mechanics.

**Theorem 8** $\chi_a(r) := 1$ *if and only if*

$$a = (\alpha, \alpha, \ldots, \alpha, \ldots), \tag{D.3}$$

*with $\alpha \in \mathbb{Q}$.*

Note that sequence (D.3) is trivially an adele since any integer $\alpha$ will be a $p$-adic integer at sufficiently large prime. A proof of this theorem is given e.g. in Chapter 3 of [55].

Sequences as the one in Eq. (D.3) (i.e. of the same integer repeated at all places) are called principal adeles, and they form a subring. Eq. (D.3) can also be thought of as a diagonal embedding of $\mathbb{Q}$ in the ring of adeles.

## D.2  Euler products

Setting $r = 1$ in Theorem 8 immediately gives the result quoted in the main text,

$$\chi_{(\infty)}(\alpha) = \prod_p \frac{1}{\chi_{(p)}(\alpha)}, \tag{D.4}$$

for any $\alpha \in \mathbb{Q}$.

Another example of a product formula, for $\alpha \in \mathbb{Q}$, is

$$|\alpha|_{(\infty)} = \prod_p \frac{1}{|\alpha|_{(p)}}. \tag{D.5}$$

This identity follows trivially from the decomposition of $\alpha$ into prime factors.

The result of Theorem 7 above can be written as

$$\mathrm{sgn}_\tau^{(\infty)} \alpha = \prod_p \mathrm{sgn}_\tau^{(p)} \alpha, \tag{D.6}$$

for all $\alpha, \tau \in \mathbb{Q}$.

For any $a \in \mathbb{Q}^\times$, the prefactors $\lambda_{(p)}(a)$ defined in Appendix B.2 obey the product formula [10]

$$\lambda_{(\infty)}(a) \prod_{p=2}^{\infty} \lambda_{(p)}(a) = 1, \tag{D.7}$$

where $\lambda_{(\infty)}(a)$ at the Archimedean place is defined as

$$\lambda_{(\infty)}(a) = \exp\left(-i\frac{\pi}{4}\,\mathrm{sgn}_{\tau=-1}^{(\infty)} a\right), \tag{D.8}$$

with $\mathrm{sgn}_{\tau=-1}^{(\infty)}$ the usual sign function on $\mathbb{R}$.

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
