# Peer review of "Building Archimedean Space"

_SciPost Physics_

## Round 2 · Referee Report · Lin Chen (Referee 1) · 2022-1-7

Report
In this manuscript, the author proposes that the theories defined over the continuum spaces can be recovered from those defined over finite places such as $p$-adic fields. And the author presents two examples to support this proposal: quantum mechanics of a free particle, and Euclidean two dimensional Einstein gravity.
In the $p$-adic quantum mechanics part, the author introduces a parameter $\tau$, and considers a quadratic extension $Q_p[\sqrt{\tau}]$. Then sign functions $sgn_\tau$ are introduced to define multiplicative characters of $Q_p^{\times}$. And the Vladimirov derivative is defined by these multiplicative characters. After these preparation, the author writes down the $p$-adic Schr$\ddot{o}$dinger equation and focuses on the free particle case. And the author finds that the position (momentum) space Archimedean propagators can be recovered by the product of the position (momentum) space $p$-adic propagators, and the Archimedean plane waves are the multiplication of $p$-adic plane waves.
In the Euclidean two dimensional Einstein gravity part, the author proposes that the Bruhat-Tits tree is the $p$-adic analogue of Euclidean $AdS_2$, and that a genus $g$ configuration of the Euclidean $AdS_2$ can be reconstructed from genus $g$ configurations of the Bruhat-Tits tree. This proposal is mainly supported by the matching of partition functions of these configurations and the comparison of the geodesic lengths in $EAdS_2$ and in the BT tree.
In this manuscript, the author also tries to construct Lorentzian $AdS_2$ by the quadratic extension of Bruhat-Tits tree. In this quadratic extension, edges are assigned spacelike and timelike labels. The author defines curvature and action on the tree, and works out the linearized Einstein equations on it.
$\textbf{Questions and comments:}$
$\textbf{Comment 1:}$
In the $p$-adic quantum mechanics part, the connection between $p$-adic theory and Archimedean theory is shown clearly. Though the results are not entirely new, the introduction of the parameter $\tau$ and sign functions $sgn_\tau$ is interesting, as can be seen from the condition $sgn_\tau(2m)=1$.
$\textbf{Question 2:}$
In the Euclidean $AdS_2$ part, the boundary points in $EAdS_2$ and in the BT tree share the same coordinate $x\in Q$. However the relation between bulk points in $EAdS_2$ and in the BT tree is still unclear. Is it possible to elaborate on this relation some more ?
$\textbf{Comment 3:}$
In the Lorentzian $AdS_2$ part, it is hard to say Lorentzian $AdS_2$ is reconstructed from this quadratic extension tree. The comparison between Lorentzian $AdS_2$ and the quadratic extension tree in section 5.6 is too rough. Important structure characterizing a Lorentzian space-time, such as causal structure, is not clear even with the assignment of space-like and time-like labels to each edge.
Though whether Lorentzian $AdS_2$ is reconstructed is doubtful, this attempt of constructing another type of tree may bring us some inspiration, and is worth being noticed.
$\textbf{Recommendation:}$
I recommend the manuscript to be published. Though the construction of Lorentzian $AdS_2$ is not so convincing, the structure of quadratic extension tree is new and may bring some inspiration. The idea of introducing a parameter $\tau$ and considering quadratic extension might turn out to be useful. It is desirable that the author elaborate some more on say causal structure of the Lorentzian tree, but this is not a necessary addition to render the paper publishable.

---

## Editorial Decision

unknown